# Large-scale crustal growth driven by LIP magmatism during the Paleoproterozoic

Matheus S. Simões[1] ✉, Andrew R. C. Kylander-Clark [2], Marcelo L. Vasquez[3], Carlos A. Sommer [4], Lucas M. M. Rossetti[5], John M. Cottle [2] & Túlio A. Mendes[3]

The evolution of Earth's continental crust is crucial for understanding geodynamics, climate regulation, and the origins of life. The Paleoproterozoic, marked by the Great Oxidation Event and the consolidation of plate tectonics, was a critical interval for continental growth. While arc magmatism dominates crust formation in the Phanerozoic, its role in earlier Earth history remains uncertain. Three silicic LIPs in the Amazon Craton were emplaced at regular ~90-100 million-year intervals (1980 Ma, 1880 Ma, and 1790 Ma), producing high-temperature (>750 °C) silicic magmas derived from lower crust (~45 km thick). Our findings demonstrate that LIPs contributed significantly to continental crustal growth through deep-crustal partial melting of Archean-Rhyacian crust. We highlight that silicic LIP magmatism was a fundamental driver of continental differentiation and long-term stability during the Paleoproterozoic.

Earth's geodynamic and continental crustal evolution is a critical issue considering its implications for climate conditions and the inception of life since the erosion of silicate rocks drives the stabilization of liquid water at Earth's surface by consuming atmospheric $CO_2$[1]. To understand this evolution, we must assess both when and how plate tectonics took place and the processes responsible for the creation of continental crust. The onset of plate tectonics as a local or global system, powered by the negative buoyancy of old dense oceanic lithosphere sinking into weaker ductile asthenosphere in subduction zones[2], dates back to Hadean[3] or the end of Archean[4]. In addition, there are two divergent hypotheses regarding the compositional evolution of the continental crust through time. A uniformitarian view based on geochemical and Hadean zircon data suggests that silicic continental crust was already formed in the Hadean, and nearly constant crustal silica compositions prevailed since the early Archean[1]. An alternative view, based on tectonic/metamorphic asymmetry, magmatic associations, and geochemistry, proposes that the early Earth was composed of a mafic crust, evolving towards more intermediate compositions from 3.1 Ga to 2.5 Ga[4].

At the beginning of the Paleoproterozoic, the Great Oxidation Event occurred during the period between 2.4 Ga and 2.0 Ga,

accompanied by decreased deposition of banded iron formation and a concomitant increase in atmospheric $O_2$ levels[5]. This time interval partially overlaps with a global decline in magmatism, orogenic activity, and passive margin sedimentation from 2.3 Ga to 2.2 Ga, interpreted as a tectono-magmatic lull[6], possibly linked to a phase of stagnant-lid tectonics[7]. The end of the Rhyacian (~2.2 Ga) to the Statherian (1.8 Ga–1.6 Ga) is characterized by multiple orogenic belts surrounding Archean cratons. Rhyacian domains, including those of the Amazon and West African cratons, are dominated by granite-basalt-komatiite associations[8,9]. Eclogites and ophiolites, considered 'plate tectonic indicators', are common features of these belts, including the Svecofennian, Kola-Karelia, and Birimian[10]. The period from the Late-Rhyacian to the Statherian includes anachronic events of rift-drift, accretionary and collisional orogenies in most Paleoproterozoic belts, with episodes of silicic additions to the continental crust[11].

An exception to these belts is the Late-Rhyacian to Statherian record of the Amazon Craton in South America. This is a key area to investigate mechanisms of crustal growth and a potential relationship with subduction, since no plate tectonic indicators have been identified. Instead, the period from 2.0 Ga to 1.74 Ga in this crustal segment is marked by the emplacement of large igneous provinces with an

[1]Caçapava do Sul Campus, Federal University of Pampa, Caçapava do Sul, RS, Brazil. [2]Department of Earth Science, University of California, Santa Barbara, CA, USA. [3]Geological Survey of Brazil, Rio de Janeiro, Brazil. [4]Department of Geodesy, Federal University of Rio Grande do Sul, Porto Alegre, RS, Brazil. [5]College of Geosciences, Federal University of Mato Grosso, Cuiabá, MT, Brazil. ✉e-mail: matheussimoes@unipampa.edu.br

extensive bimodal igneous association[12–14]. Peaks in magmatic activity within these provinces occurred at approximately 90 to 100 Myr intervals, specifically around 1.98 Ga, 1.88 Ga, and 1.79 Ga, accompanied by the emplacement of basaltic radial dyke swarms and sill complexes[15]. These silicic LIPs are key to unraveling the tectonic evolution of the Amazon Craton, evaluating effective mechanisms of continental crust formation, and testing the theory of a global interconnected network of plate boundaries during the Paleoproterozoic.

The Amazon Craton spans approximately 6,500,000 km², making it the largest continuous exposure of Archean, Paleoproterozoic, and Mesoproterozoic crust in South America. It is bounded by the Andes and partially concealed by Andean foreland basins to the west and Neoproterozoic mobile belts to the east. The craton is divided into the Guyana Shield in the north and the Central Brazil Shield in the south, separated by the overlying Phanerozoic Amazonas Basin[16]. Paleomagnetic reconstructions indicate a connection between the Amazon and West African cratons around -2.0 Ga, following the 2.2–2.0 Ga Transamazon-Birimian Orogeny[17]. At 1.78 Ga, paleomagnetic models suggest a connection between Amazonia, West Africa, and Baltica, forming the South America-Baltica (SAMBA)[18] configuration (Fig. 1A). The SAMBA model is considered the core of the Columbia Supercontinent, with other cratonic blocks likely accreted to this extensive continental landmass, even though some researchers dispute that Amazonia/West Africa was never part of Columbia[19].

The geological framework of the Amazon Craton consists of Archean provinces considered the craton's central nuclei, represented by the Central Amazonian Province (>2.3 Ga), bounded by progressively younger Paleoproterozoic provinces[20] (Fig. 1B). Archean segments (Mesoarchean to Neoarchean) outcrop in the eastern portion of the Central Brazil Shield, and at the east and west boundaries of the Guyana Shield[20]. Archean crust is indirectly observed beneath most of the Central Brazil Shield[21] and part of the Guyana Shield. The distribution of Archean crust is evidenced as an isotopic provinciality that divides the Amazon Craton into an eastern segment, characterized by Archean Nd depleted mantle model (TDM) ages and the most negative $\varepsilon Nd_{(t)}$ values, and a western segment, where Rhyacian to Orosirian Nd TDM ages and slightly negative to slightly positive $\varepsilon Nd_{(t)}$ values predominate (Fig. 2A, B).

Archean provinces in the Amazon and West African cratons are bounded by high-grade metamorphic belts, greenstone belts, tonalite-trondhjemite-granodiorite (TTG) to charnockite suites, and high-Mg-K granite suites from the Rhyacian to Orosirian (2.2–1.95 Ga). These form a 1500 km-long NNW-SSE belt in the northern Guyana Shield and occur as several fragments in the Central Brazil Shield[22]. This orogenic event, widespread across both the Amazon and West African cratons, is related to the Transamazonian Orogeny in South America. Orosirian to Stenian provinces have been interpreted as NW-trending orogenic belts characterized by continuous arc accretion and soft-collision

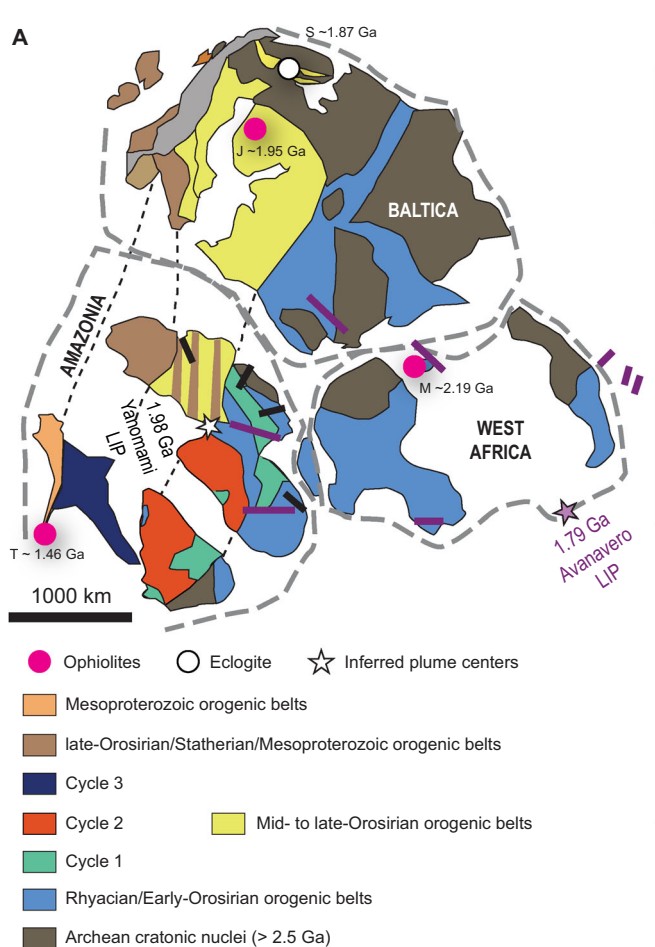

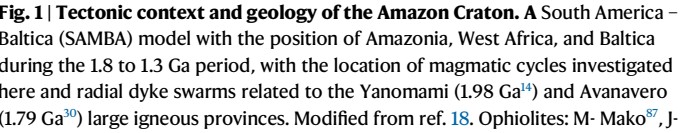

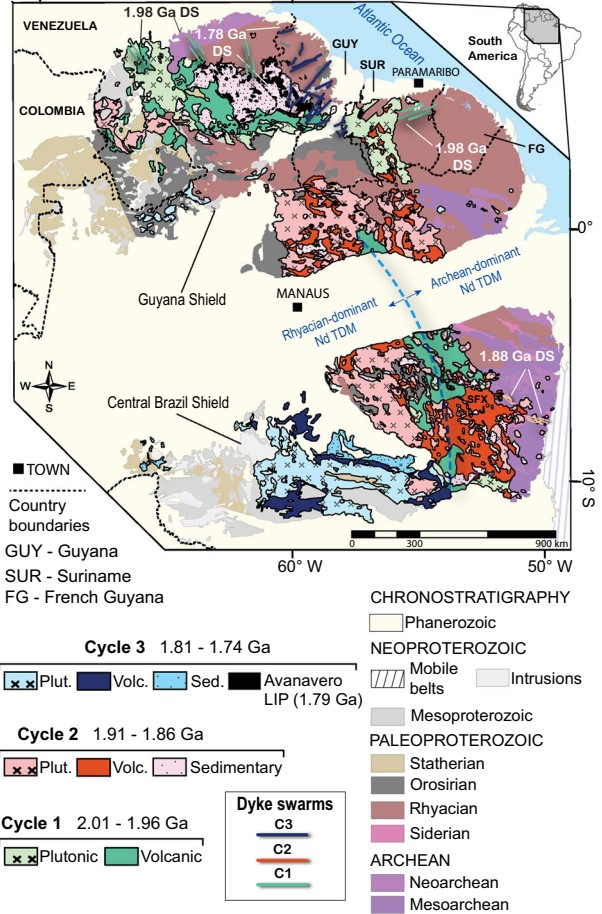

**Fig. 1 | Tectonic context and geology of the Amazon Craton. A** South America – Baltica (SAMBA) model with the position of Amazonia, West Africa, and Baltica during the 1.8 to 1.3 Ga period, with the location of magmatic cycles investigated here and radial dyke swarms related to the Yanomami (1.98 Ga[14]) and Avanavero (1.79 Ga[30]) large igneous provinces. Modified from ref. 18. Ophiolites: M- Mako[87], J- Jormua[88] and T- Trincheira[24]. Eclogite: S- Salma[89]. **B** Chronostratigraphic map of the Amazon Craton (the southern Paraguá and Rio Apa domains are not shown in this figure, as their inclusion would alter the map scale and reduce the detail of the LIP–SLIP units and sample locations). The magmatic cycles indicated in the legend are interpreted in this work. Modified from ref. 22.

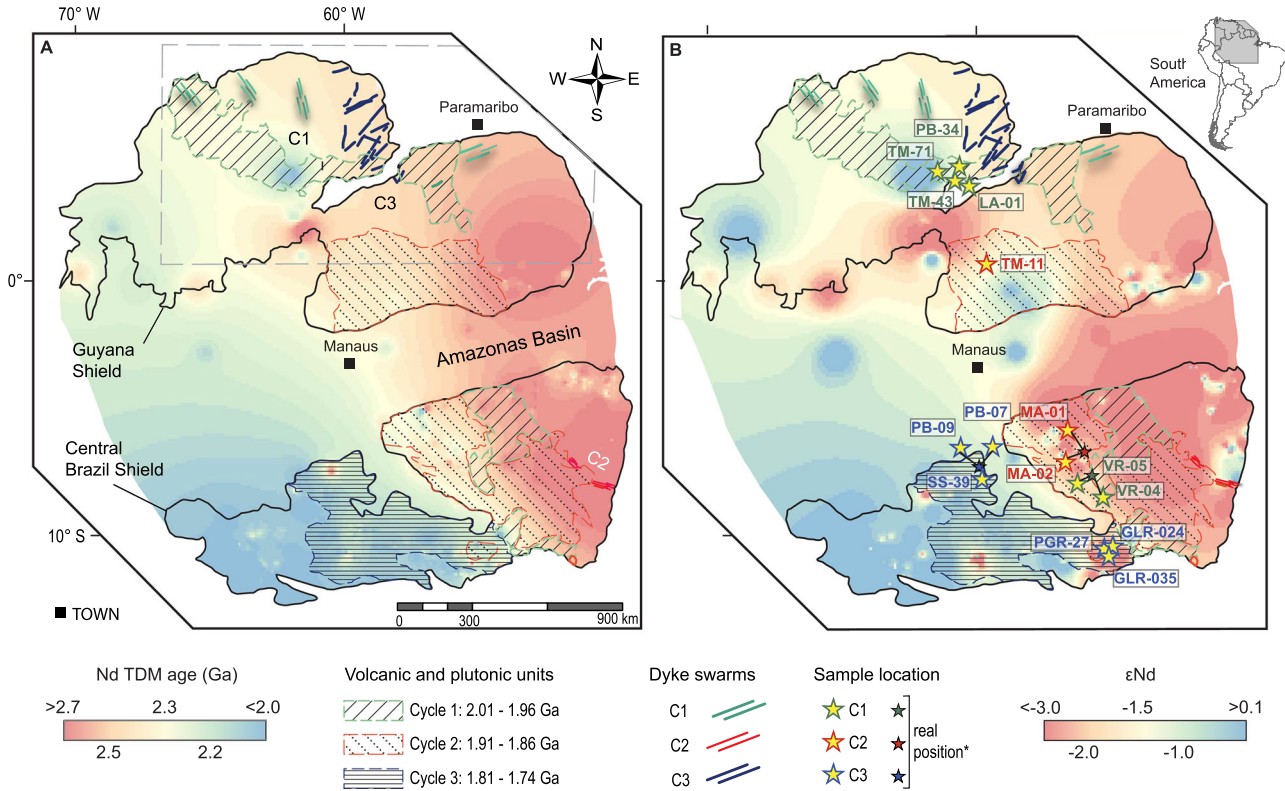

**Fig. 2 | Contour maps illustrating interpolated Nd isotope data and the distribution of silicic LIPs across the Amazon Craton.** The maps were built using the IDW interpolation tool of the QGIS 3.38.1 software with P coefficient distance of 2 and a quartile mode of linear interpolation. **A** Interpolation of Nd TDM second-stage ages, highlighting the division of the Amazon Craton into an Archean-dominated crust in the east and a Rhyacian to Orosirian-dominated crust in the west. The dashed line rectangle corresponds to the area of Fig. 4. **B** Interpolation of

εNd values, displaying isotopically juvenile domains to the west and more evolved domains to the east. The database for contour construction is sourced from refs. 21 and 41. Boundaries of the Guyana and Central Brazil shields and the silicic LIPs are adapted from ref. 22. Dyke swarms are mapped following refs. 14, 26, 30. Sample locations are symbolized by big yellow stars. Small stars indicate clusters of samples and represent their real position. Isotope data compilation is present in Supplementary Data 1 for reproducibility.

events to the west of the Archean-Rhyacian cores at 2.03 Ga–1.87 Ga, 1.83 Ga–1.52 Ga, 1.55 Ga–1.3 Ga, and 1.3 Ga–1.0 Ga[16].

Parallel accretionary orogens were matched with orogens from Baltica and Laurentia, and paleomagnetic correlations linked them with Amazonia, giving rise to the Great Proterozoic Accretionary Orogen, thought to be generated by almost continuous subduction over time[23]. Nevertheless, the record of plate tectonic indicators in the Amazon Craton comprises a 1460–1440 Ma ophiolite[24] in the Sunsás Belt, that post-dates Paleoproterozoic belts and pre-dates the late Mesoproterozoic events (1.2–0.95 Ga). These events are correlated with the Grenville-type orogens related to the collision of Amazonia and Laurentia[25].

Large Paleoproterozoic linear volcano-plutonic belts across the Amazon Craton are composed of silicic-dominant igneous rocks, mostly granitoids and rhyolitic ignimbrites with calc-alkaline I-type to A-type affinities, that intrude and overlie Archean, Rhyacian, and Eo-Orosirian terranes (Fig. 1B). They have been previously interpreted as part of distinct magmatic arcs or related to post-collisional events, conforming with the orogenic evolution of the Amazon Craton provinces during the Paleoproterozoic[16]. The main arguments favoring magmatic arc settings involve: (i) geochemical features, such as the occurrence of high-K calc-alkaline rocks with negative Nb-Ta anomalies, (ii) slightly juvenile Nd isotope signatures, (iii) genetic association with Au-Cu(-Mo) and Cu-Mo epithermal and sparse porphyry mineralization, and (iv) regional crystallization/Nd TDM age zonation. All these features are also likely to occur in LIP-SLIP settings, and several other features from these belts, such as radial dyke swarms, high-volume igneous events with short duration, bi-modal compositions,

and older surrounding crust, are typical of intraplate LIP magmatism (Supplementary Data 2). Further, three separate LIP/Silicic LIP events have been interpreted, based on their preserved areal extent, the short duration of igneous activity, and correlation with tholeiitic basaltic radial dyke swarms[15]

1) The 'early Orosirian' Orocaima Silicic LIP[15] in the central part of the Guyana Shield has more than 0.4 Mkm², an approximate duration of ~53 Myr, and peak magmatism at 1.98 Ga[14,15], representing the first cycle of large-scale silicic magmatism, referred to here as Cycle 1. This Silicic LIP occurs spatially and temporally close to the Yanomami LIP, represented by 1.98 Ga radiating dyke swarms, which are composed of tholeiitic basalts derived from an OIB-like primary melts and form an arrangement pointing to a plume center near the SW margin of proto-Amazonia[14] (Fig. 1A);

2) The 'late Orosirian' Uatumã Silicic LIP[12] that occurs over Archean terranes and the Transamazonian belts along the Guyana and Central Brazil shields, has nearly 1 Mkm², an approximate duration of ~47 Myr, with peak magmatism at 1.88 Ga[12], representing the second cycle of large-scale magmatism at around 1.88 Ga (Cycle 2). The silicic magmatism is locally associated with charnockites, enderbites, rapakivi granites and gabbros, being also recorded in NW-trending bimodal dyke swarms[26] (Fig. 1B). The age of this magmatism correlates with worldwide basaltic dyke swarms associated with several LIPs[27];

3) The Statherian Western Amazonia Igneous Belt[13], which occurs only within the Central Brazil Shield, generally intrudes the Rhyacian to Orosirian basement, with at least 0.4 Mkm², a peak of magmatism at 1.78 Ga, and a duration between 45 and 50 Myr. The

SLIP is associated with high- and low-Ti tholeiitic basaltic dyke swarms[28] in the Central Brazil Shield and is synchronous with the 1.79 Ga Avanavero LIP in the Guyana Shield[29]. This LIP is represented by a sill complex in the Guyana Shield and by radiating dyke swarms in both Amazon and West African cratons, composed of E-MORB-like tholeiitic high- and low-Ti basalts whose arrangement points to a plume center in the Tuareg Shield[30] (Fig. 1A). This represents the third cycle of large-scale magmatism (Cycle 3).

In this study, we analyze zircon U-Pb, Lu-Hf, and trace element data from representative volcanic rocks across each of these Large Igneous Provinces to interpret the geodynamic processes driving the repeated cyclic eruptions of voluminous magmas during the Paleoproterozoic. Our findings are compared to the crustal evolution of the Amazon Craton, serving as an example of mechanisms responsible for large-scale continental crust growth.

## Results

### Age of volcanic rocks

To determine crystallization and inherited age ranges along with zircon trace element compositions, we analyzed 15 samples representative of Paleoproterozoic LIPs from the Amazon Craton. Field and petrographic description, along with location in more detailed geological maps, are present in Supplementary Data 3. The analyses were conducted on 370 previously imaged zircon grains (Supplementary Data 4). Sample selection criteria focused on key lithostratigraphic units with well-established geological contexts and well-characterized whole-rock geochemistry. Crystallization ages were obtained by weighted mean of $^{207}Pb/^{206}Pb$ dates. Ages interpreted as inherited are reported as concordant (>95%) zircon $^{207}Pb/^{206}Pb$ apparent ages of single grains. The complete dataset is presented in the Supplementary Data 5.

Six samples of Cycle 1 were collected along the northern boundary of Brazil and Venezuela in the Guyana Shield (three samples from the I-type Surumu Group and one sample from the A-type Cachoeira da Ilha Formation) and near the Tapajós Gold Province in the Central Brazil Shield (two samples from the I-type Vila Riozinho Formation). The three samples from the Surumu Group yield crystallization ages of 1987 ± 4 Ma, 1995 ± 5 | 8 Ma, and 1996 ± 4 Ma, with inherited ages ranging from 2307 Ma to 2021 Ma. The sample from the Cachoeira da Ilha Formation produces a crystallization age of 2011 ± 4 Ma, with inherited ages between 2097 Ma and 2061 Ma. The two samples from the Vila Riozinho Formation yield ages of 1989 ± 4 Ma and 1982 ± 3 Ma, with inherited ages ranging from 2049 Ma to 2026 Ma.

Three samples from Cycle 2 were collected from both the Guyana Shield (one sample from the I-type Jatapu Formation) and the Central Brazil Shield (two samples from the A-type Moraes de Almeida Formation). The sample from the Jatapu Formation yields an age of 1890 ± 4 | 8 Ma, with inherited grains dating from 2135 Ma to 1976 Ma. The Moraes de Almeida Formation samples produced ages of 1882 ± 6 Ma and 1893 ± 4 | 5 Ma, with inherited ages ranging from 2042 Ma to 1922 Ma.

Six samples from Cycle 3 were obtained from the lowermost (four samples from the A-type Colíder Group) and uppermost (two samples from the A-type Pedro Sara Formation) stratigraphic units of the Western Amazonia Igneous Belt. The Colíder Group samples yield ages of 1814 ± 3 Ma, 1796 ± 4 | 7 Ma, 1795 ± 5 Ma, and 1792 ± 4 Ma, with inherited ages ranging from 1971 Ma to 1829 Ma. Two samples from the Pedro Sara Formation provided ages of 1762 ± 4 Ma and 1755 ± 4 Ma, with inherited grains ranging from 2852 Ma to 1810 Ma.

### Zircon composition, crystallization conditions, and source-depth

Zircons from all three cycles exhibit similar compositions, characterized by high concentrations of Hf, Nb, Ta, Y, Th, and P, consistent with those of granitoids and dolerites[31]. Their mean Th/U and Yb/Sm are approximately 1 and 20, respectively. Chondrite-normalized trace element patterns reveal negative anomalies for La, Pr, Eu, Nb, and Ti, coupled with positive anomalies for Th, U, Hf, and Ta. The mean rare earth element (REE) content of the zircons is ~1000 ppm, with chondrite-normalized REE patterns displaying positive Ce and Sm anomalies and a notable enrichment of heavy REE compared to light REE (Supplementary Data 6).

Trace element geochemistry of zircons provides insights into their source rock composition and crystallization environment[31]. Zircons formed from garnet-bearing sources exhibit lower heavy REE concentrations due to garnet's affinity for these elements during partial melting, resulting in flatter chondrite-normalized heavy REE patterns[32]. Compared to zircons from high-temperature, high-pressure metamorphic rocks equilibrated with garnet, the igneous zircons analyzed here display $Yb_{(n)}$, $Yb/Ce_{(n)}$, and $Yb/Tb_{(n)}$ similar to those of lower crustal material[33](Fig. 3A, B). This source interpretation is consistent with field evidence indicating that melts for cycles 1 and 3 were generated from HT-UHT garnet-bearing migmatites and granulites at lower continental crustal levels[34,35].

To estimate crustal thickness for each studied magmatic cycle, we applied the chemical mohometry model[36], by using a large dataset of geochemical data available from the literature. Data were filtered to exclude major and mobile trace elements sensitive to hydrothermal alteration processes. The results yielded Moho depths at 46.3 km during Cycle 1, 46.4 km during Cycle 2, and 43.6 km during Cycle 3 (Supplementary Data 7), which corresponds to a 'normal' continental crust thickness, close to the ~40 km average[37]. The depth of zircon crystallization was estimated by the model utilizing the $^{176}Lu/^{177}Hf$ ratio of individual grains as a pressure proxy[38]. Given the inherent uncertainty of single-zircon pressure determinations, Kernel Density Estimate (KDE) curves were used for more robust analysis. The KDE plots reveal peaks at approximately 8.4 kbar for Cycles 1 and 2, and 7.4 kbar for Cycle 3 (Fig. 3C). Assuming a pressure gradient of 3.5 kbar/km, these pressures corresponded to crystallization depths of ~29 km for Cycles 1 and 2 and ~27 km for Cycle 3, consistent with field evidence and zircon trace-element data indicating sources within the lower continental crust. Calculated Ti-in-zircon crystallization temperatures[39] range from 650 °C to 950 °C, with most samples clustering between 780 °C and 820 °C. Using zircon-melt partitioning of cerium[40], $fO_2$ conditions for single zircon grain crystallization were estimated. Results indicate a range from reducing conditions at ΔQFM −2 to oxidizing conditions up to ΔQFM + 7, correlating with mineral assemblages that include biotite ± amphibole ± Fe-Ti oxides. Cycle 1 shows a progressive transition from more oxidizing conditions before 2000 Ma to more reducing conditions afterward. Cycle 2 spans $fO_2$ conditions between ΔQFM + 3 and −3, with no clear age-related trend. Cycle 3 zircons (1810–1780 Ma), primarily associated with biotite ± amphibole ± Fe-Ti oxide-bearing rocks, formed under dominantly oxidizing conditions. In contrast, zircons from the youngest Pedro Sara Formation (~1760–1755 Ma), composed of clinopyroxene ± orthopyroxene ± Fe-Ti oxides, crystallized under more reducing conditions (Fig. 3D).

### Hf isotopic signatures from Paleoarchean to Tonian in the Amazon Craton

In addition to previously published datasets including detrital and igneous zircon age and Hf isotopes[41], we present 172 coupled zircon U-Pb and Lu-Hf analyses from 15 volcanic rock samples representing the three Paleoproterozoic igneous cycles. Reference values for the chondritic uniform reservoir (CHUR) are from[42], whereas those for the depleted mantle (DM) are from[43]. Evolution lines for a typical Archean mafic source and products of TTG remelting were anchored at 2200 Ma, representing the major episode of crust recycling during the

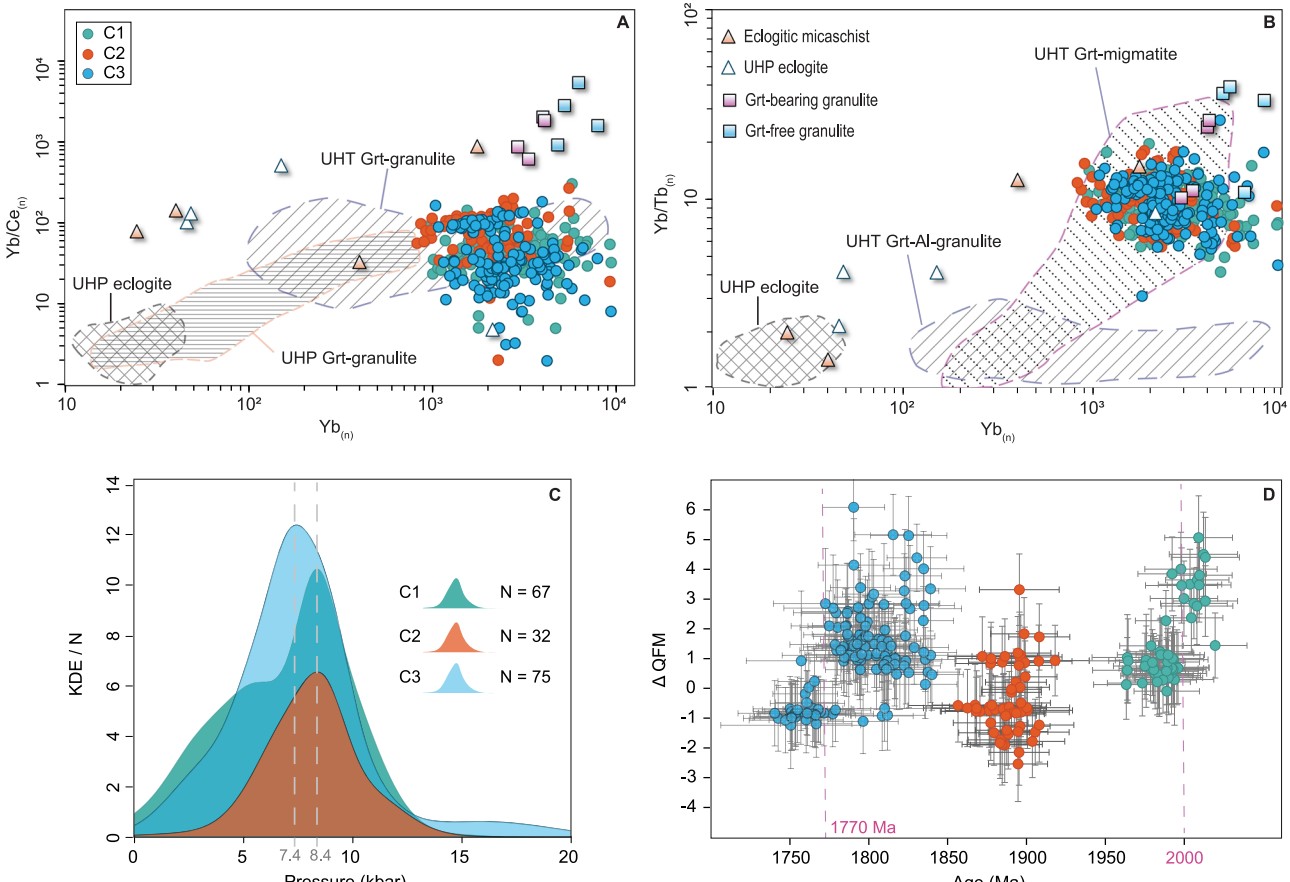

**Fig. 3 | Crystallization conditions and trace-element composition of zircons.**
**A**, **B** Plot of chondrite-normalized[90] rare-earth elements Yb, Ce and Tb analyzed for our samples and compared to metamorphic zircons from eclogitic metasediment (Eclogitic micaschist), ultrahigh-pressure eclogites and metasediments (UHP eclogite), garnet(grt)-bearing granulites and garnet(grt)-free granulites[32]. The fields correspond to additional analyses of zircons from ultrahigh-pressure eclogites (UHP eclogite[91]), ultrahigh-pressure garnet-bearing felsic to intermediate granulites (UHP Grt-granulite[92]), ultrahigh-temperature garnet-bearing aluminous granulite (UHT Grt-granulite[93]), and garnet-bearing aluminous granulite and anorthosite[94]. The composition of zircons from our samples indicates affinity with partial melts

from garnet-bearing high-temperature crustal sources. Data for the reproduction of literature fields are present in Supplementary Data 8. **C** KDEs of crystallization pressure estimates using the $^{176}Lu/^{177}Hf$ ratio of individual grains as a pressure proxy[38]. **D** Oxygen fugacity for individual zircon grains expressed as ΔQFM units along the quartz-fayalite-magnetite buffer calculated using the zircon-melt partitioning of cerium model[40] (errors are in 1σ). The pink dashed lines depict the ages (2000 Ma and 1770 Ma) when the transition from more oxidizing to more reducing conditions took place during cycles 1 and 3. Trace-element, Ti-in-zircon, and oxygen fugacity results are present in the Supplementary Data 5.

Transamazonian Orogeny[44], and at 3500 Ma, close to the oldest $Nd_{TDM}$ age found for Cycle 2 igneous rocks[45] and the oldest zircon in the literature dataset[41], representing the oldest hypothetical recycled crustal source. Literature data for each Cycle represent primarily intrusive granitoids along with a few volcanic samples. The combined analysis of our data and published datasets reveals distinct trends in Hf isotope compositions for each cycle (Fig. 4).

Cycle 1 zircons exhibit a range from supra-chondritic[42] radiogenic signatures, with εHf(t) values reaching +10 and close to the Depleted Mantle (DM) evolution curve[46], to less radiogenic signatures with εHf(t) values as low as −12, as observed in the εNd vs. U-Pb age space. Our data, obtained in samples from the Guyana Shield agree with data obtained for volcanic and plutonic rocks of the same age from the Central Brazil Shield. Primary sub-populations for Cycle 1, identified in KDE plots, display peaks at −5.4 and +2.7. Cycle 2 zircons display a bimodal distribution of εHf(t) values. One subset shows radiogenic signatures, clustering near −0.9 and reaching up to +5, and another subset has predominantly unradiogenic signatures, with εHf(t) values clustering around −15 and reaching as low as −20. The most unradiogenic signatures were obtained for zircons of Cycle 2-related granitoids intruding the Archean provinces of the AC. Granitoids intruding Archean crustal segments within the eastern

Amazon Craton display strongly negative εHf(t), εNd(t), and Archean $Nd_{TDM}$ ages[47], mirroring the most negative εHf(t) values of Fig. 4, suggesting the origin of the least radiogenic Hf values linked to significant Archean crustal reworking. They also follow the evolution of Archean crust, as observed in detrital zircon sets obtained for Amazon River sands and Paleoproterozoic sedimentary units[41]. Cycle 3 zircons from this study and from the literature range from slightly radiogenic signatures, with a peak at +0.68, with εHf(t) values reaching +5 (resembling the behavior of εNd for literature data of this Cycle), to values as low as −10.

## Discussion

### Isotope evolution of the CA during intraplate magmatism

The tectonic evolution of the Amazon Craton begins in the Mesoarchean, based on surface geological data, or possibly the Hadean to Paleoarchean, considering zircon xenocrysts and $Nd_{TDM}$ ages. The oldest known zircon grain, dated at 4219 ± 19 Ma, was found in ~1980 Ma volcanic rocks from Cycle 1[48], implying that younger crust was built on a Hadean substrate. Igneous and detrital zircons from the end of Archean are characterized by near-chondritic Hf and Nd isotopic signatures, with evolved rocks plotting between the evolutionary trends of Archean basalt C-F2[49,50] and TTG melting,

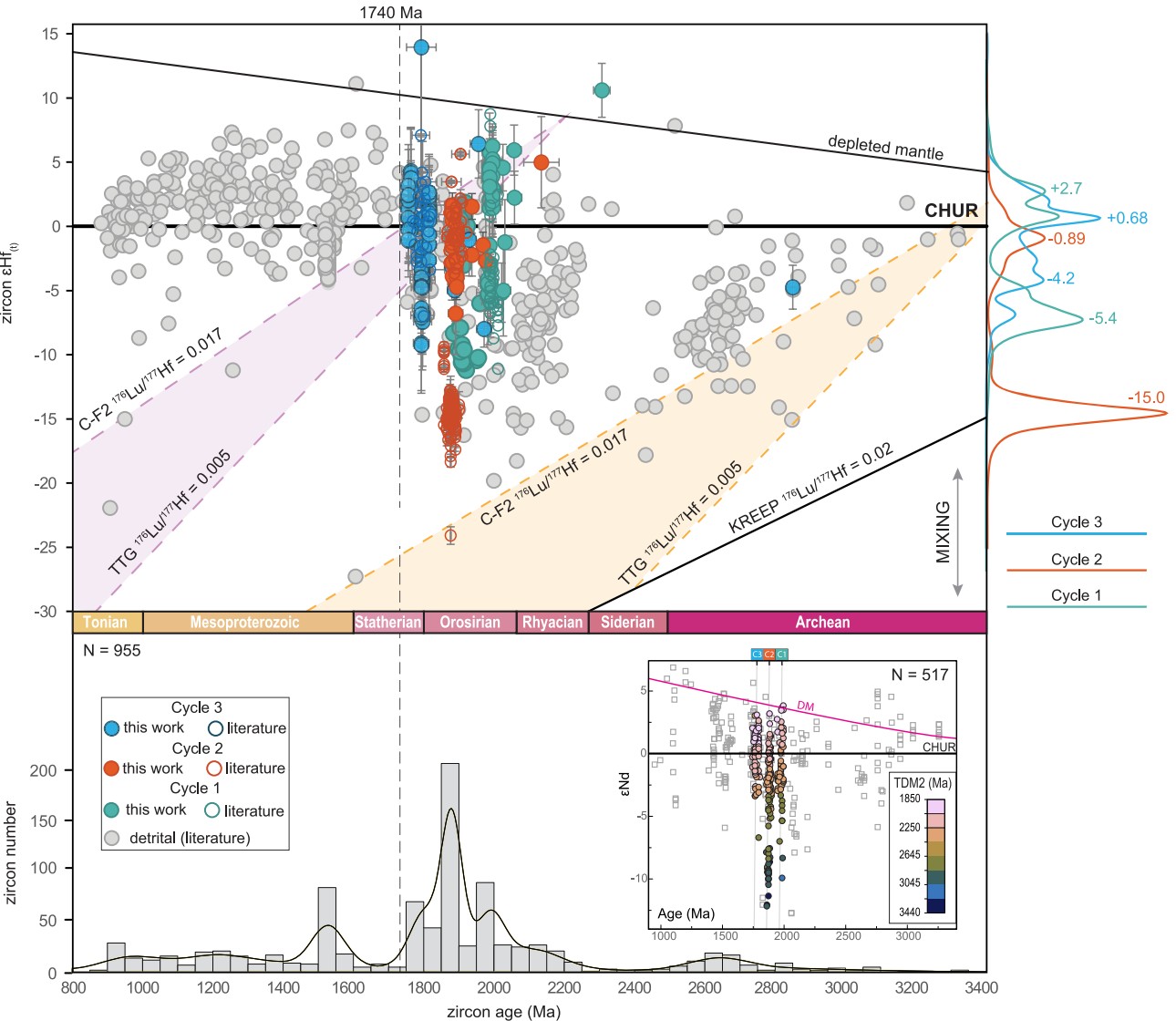

**Fig. 4 | Plot of initial epsilon hafnium (εHf$_{(t)}$) versus zircon age for the three silicic LIPs studied here and Archean to Tonian detrital and igneous zircons from the Amazon Craton. Error bars are at 2σ level.** KREEP is the Hadean incompatible element enriched lunar basalt ($^{176}$Lu/$^{177}$Hf = 0.020[50]), CF-2 is an Archean mafic source ($^{176}$Lu/$^{177}$Hf = 0.017[49]), and TTG is a tonalite-trondhjemite-granodiorite source ($^{176}$Lu/$^{177}$Hf = 0.005[50]). The depleted mantle (DM) curve is plotted using the values of ref. 46. The kernel density estimate (KDE) curve in the horizontal axis was built with new and previously published zircon ages. The KDE in the vertical axis comprises new Lu-Hf data from the silicic LIPs, displaying the most frequent εHf values. The inset depicts literature Nd isotope data for the Amazon Craton in the εNd against U-Pb age space, compiled from the same sources of Fig. 2. Circles are data from igneous rocks from the three igneous cycles and are color-coded according to their second-stage TDM ages. Unfilled squares are data from rocks that do not belong to the igneous cycles. DM curve according to [57]. Data for the reproduction of this figure are in Supplementary Data 1 and 9.

as well as mixed compositions between these endmembers. Few zircons exhibit Siderian ages, consistent with the global tectono-magmatic lull from 2.4 Ga to 2.3 Ga[6]. During the Rhyacian to Orosirian, a renewed period of zircon production occurred, with compositions ranging from supra-chondritic (near DM) to near-chondritic Hf-Nd signatures. This interval is marked by mixing trends with unradiogenic crust, reaching εHf$_{(t)}$ values as low as −20, which document a prolonged isotopic pulldown from the Paleoarchean through the Orosirian.

This isotopic pulldown is marked by increasing Nd$_{TDM}$ ages and reflects two main episodes of Paleoarchean crustal reworking: one during the late-Archean and another during the Rhyacian to Orosirian. Rhyacian rocks in the Amazon Craton are primarily associated with granite-greenstone belts containing basalts and TTG rocks[8], whereas Orosirian magmatism is dominated by bimodal, silicic plutonic and volcanic rocks[15]. After this isotopic pulldown, Statherian zircons exhibit predominantly intermediate εHf$_{(t)}$ values between +5 and −5, with a few reaching −10 to −15. Despite these higher εHf$_{(t)}$ values, inherited zircons and Nd$_{TDM}$ ages suggest Archean crustal reworking persisted during the Statherian. From ~1740 Ma, the age marking the end of Cycle 3, through the Tonian, most zircons display supra-chondritic radiogenic Hf signatures, also observed for whole-rock Nd isotopes. A smaller subset shows εHf$_{(t)}$ values between 0 and −10, with only a few Tonian zircons reaching −10 to −25. The least radiogenic Tonian zircons plot within the evolutionary trends of Rhyacian C-F2 and TTG, indicating a potential crustal source of this age with minimal Archean crustal reworking. This shift towards supra-chondritic zircon isotope compositions after ~1740 Ma aligns with the predominance of Mesoproterozoic mantle-derived lithologies across the Amazon Craton, including gabbros, charnockites, anorthosites, and A-type granitoids associated with Anorthosite-Mangerite-Charnockite-Granite (AMCG) suites[22].

## Changes in the structure and composition of the Amazon Craton crust

The Amazon Craton displays significant structural and compositional contrasts across different provinces and time periods. In the Archean provinces of the eastern Central Brazil Shield, between 3.0 Ga and 2.6 Ga, the crust evolved from dome-and-keel structures—characterized by TTG suites surrounded by greenstone belts—to linear belts consisting of gneisses and syn-tectonic granitoids. This transformation has been interpreted as marking the onset or transition to modern-style plate tectonics[51]. In the Guyana Shield, analogous dome-and-keel structures, dated between 2.18 Ga and 2.13 Ga, were later reworked during the Transamazonian Orogeny (2.11–2.02 Ga[8]). By this time, the exposed Rhyacian crust consisted of approximately 75% TTG rocks with intermediate compositions, accompanied by greenstone belts with lithologies ranging from ultramafic to silicic. Following the Transamazonian Orogeny, the crust underwent significant changes, including ultrahigh-temperature metamorphism and gabbro-charnockite magmatism[8]. This was followed by the emplacement of the Yanomami LIP[14] and large-volume silicic igneous provinces, marking a pronounced shift towards a crust dominated by silicic compositions. This transition is evidenced by the widespread intrusion and extrusion of silicic igneous rocks and the deposition of siliciclastic continental sediments (Fig. 5).

The extensive silicic magmatism from the Orosirian to the Statherian is defined by bimodal compositions, the coexistence of A- and I-type magmas, contemporaneity with basaltic radial dyke swarms and sill complexes, short-lived times of duration, and evidence of coeval crustal anatexis. These features suggest an intraplate tectonic setting for this period of significant continental crust growth and differentiation[12–14]. Hot silicic magmas (>750 °C) formed through partial melting of the lower crust in response to repeated emplacement of mantle-derived tholeiitic basaltic magmas. Zircon crystallization data constrain these melts to the lower continental crust, where they originated from high-grade crustal rocks before ascending to the upper crust (Fig. 6A–C). Crystallization occurred under variable redox conditions, with no evidence of voluminous S-type magmas.

As observed in other LIP-related settings[52], the isotope composition of silicic igneous rocks reveals that the proportion of reworked crust during magmatism was highly dependent on the availability of pre-existing crustal structure and age. These magmas represent partial melts of a heterogeneous crust, dominated by Archean material near Archean outcropping segments in the eastern Amazon Craton. This is reflected in isotopic signatures, such as unradiogenic Nd-Hf values and Archean Nd TDM ages for Cycles 1 and 2. Toward the western portion of the craton, the crust transitions from an Archean-dominated composition to Rhyacian crust, with contributions of more radiogenic Nd-Hf signatures and Rhyacian Nd TDM ages, although locally truncated by Archean segments (Figs. 2, 6), as recorded by igneous rocks from cycles 2 and 3. Cycle 3 took place within a distinct crustal segment, with a younger basement (2.2–1.85 Ga) locally presenting outcropping Archean rocks along with Archean Nd TDM and inherited zircon ages. Each subsequent igneous cycle also involved the melting of lower-crustal materials associated with earlier LIP events. For instance, Cycle 2 rocks show Archean and Rhyacian Nd TDM ages, but also ages corresponding to Cycle 1. This is further supported by inherited zircon data from each cycle, indicating the assimilation of earlier LIP-related rocks during magmatism.

Following the emplacement of these magmatic provinces, the Paleoproterozoic crust of the Amazon Craton evolved into a layered structure. The upper crust became dominated by extensive silicic volcanic and sedimentary strata, while the mid-crust comprised low- to medium-grade metamorphic rocks intruded by granitoids. Exhumed lower crustal segments contained mafic to aluminous granulites and charnockites, ranging from garnet-free to garnet-bearing lithologies[8,34,35]. This crustal architecture is comparable to the composition of present-day continental crust and its structural arrangement[37] (Fig. 6D).

## Repeated cycles of mantle-derived magmatism at ~100–90 Myr Intervals

The intraplate magmatism in the Amazon Craton occurred at consistent, regular intervals. Basaltic dyke swarms associated with Cycles 1, 2, and 3 have U-Pb zircon and baddeleyite ages of 1980 Ma[15], 1880 Ma[26], and 1790 Ma[30], respectively, pointing to 100 Myr and 90 Myr intervals that potentially represent episodes of mantle partial melting. Contemporaneous silicic rocks yield zircon-age peaks at 1990 Ma, 1886 Ma, and 1789 Ma, as shown in KDE plots (Fig. 7A), corresponding to 104 Myr and 97 Myr intervals, representing peaks of zircon production by crustal partial melting. These intervals closely match harmonic mantle cycles of ~93.5 Myr, identified through time-series analysis of global datasets, including U-Pb ages, Lu-Hf and Re-Os isotopes, and LIP ages[53]. The 93.5 Myr and 187 Myr mantle cycles have persisted through at least 85% of Earth's history and are correlated with the terminal ages of superchrons[41]. As with LIP events in the Amazon Craton, no consistent relationship exists between these cycles and the assembly or breakup of supercontinents or collisional orogens[53].

Mantle-related magmatic cycles and LIP emplacement are not unique to the Amazon Craton. Numerous 'barcode' correlations between Paleoproterozoic LIPs suggest linked magmatic events across connected landmasses[15]. For example, basaltic dyke swarms and sills dated at 1.98–1.95 Ga within the Kola Craton[54] are contemporaneous with Cycle 1. Similarly, 1.88–1.86 Ga LIPs in the Siberian, North China, Bastar and Superior cratons correspond to Cycle 2[27]. Cycle 3 magmatism is reflected in ~1790-1750 Ma bimodal dyke swarms and sill complexes with intraplate geochemical signatures within the West African Craton. These events also correlate with intraplate magmatism in northwestern Laurentia and Siberia[55]. These global mantle-driven LIP events, as exemplified by the evolutionary record in the Amazon Craton, highlight the importance of LIP magmatism as a complementary mechanism to arc magmatism in driving crustal growth during the Paleoproterozoic.

## Crustal growth by intraplate igneous events during the Paleoproterozoic

Based on the geological, geochemical, and isotopic data presented here, major mechanisms of continental crust growth - particularly during the Paleoproterozoic - are compared and discussed. The most widely accepted processes for continental crust formation include oceanic plateau formation, arc-accretion, and calc-alkaline magmatism[56]. This model arises from the explanation of the average dioritic-to-tonalitic composition of continental crust, attributed to processes such as basalt fractional crystallization, melting of subducted or underplated mafic lithosphere, and fractionation of high-Mg andesites, all of which require the presence of water[57]. The dominance of subduction-related processes in continental crust formation is inferred from typical geochemical signatures, such as negative Nb and Ta mantle-normalized anomalies and excess Pb content. These features are considered hallmark indicators of arc magmas[56]. Consequently, models of crustal growth that emphasize arc magmatism inherently assume a major role for plate tectonics and subduction processes. As[56] aptly summarized: "No water, no oceans, no plate tectonics, no granites, no continents."

Nonetheless, the mantle transition zone is a major $H_2O$ reservoir[58], and hydrous upwellings from this region may generate intraplate volcanism[59]. Large volumes of silicic magmas, however, can only be produced predominantly through partial melting of fertile, hydrous lower-crustal material previously generated by subduction[60]. Such processes not only facilitate the generation of extensive silicic magmatism but also account for mantle-normalized negative Nb and Ta anomalies, together with elevated Pb concentrations. Such crustal

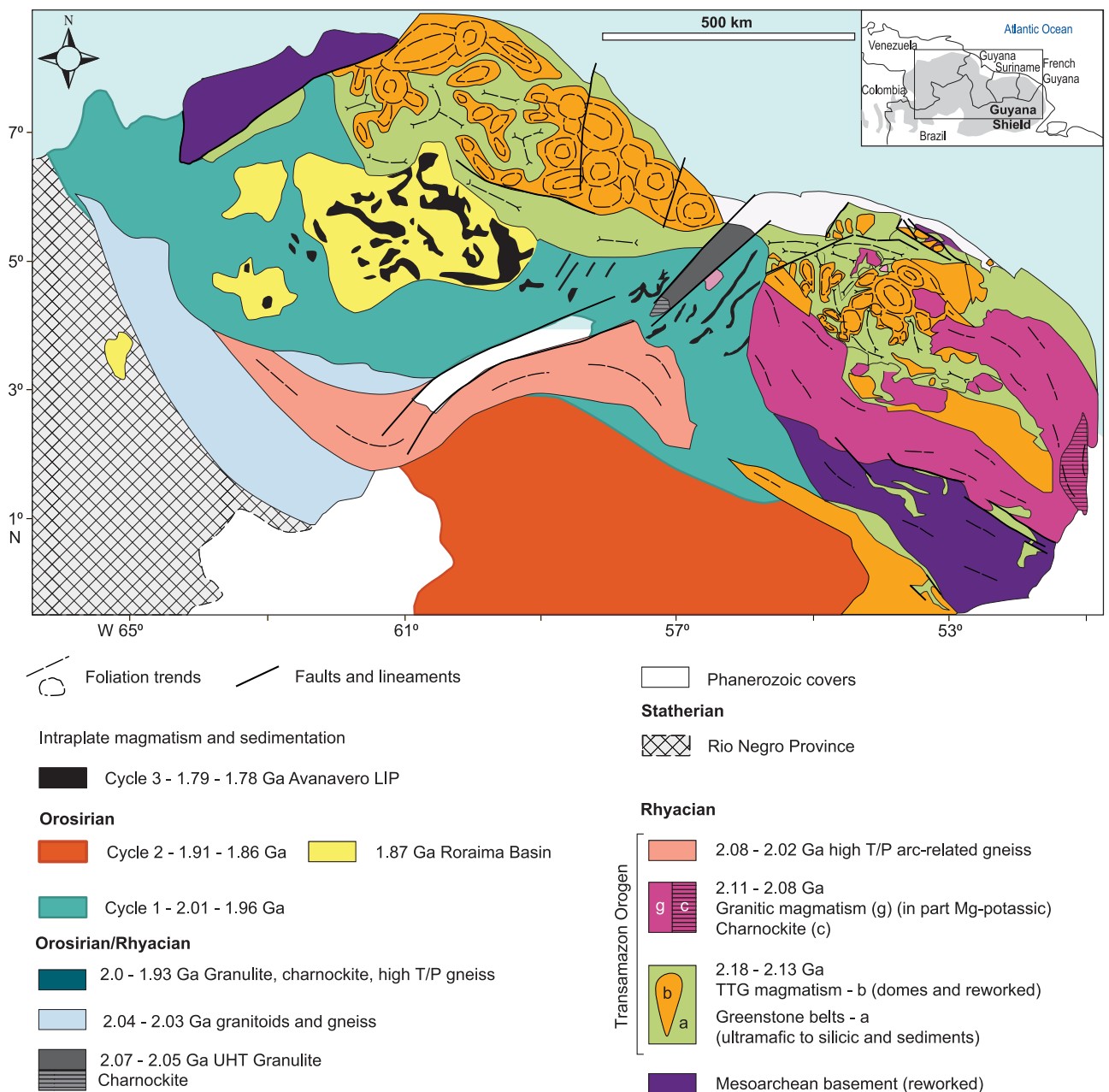

**Fig. 5 | Geological map with structural sketch of the Guyana Shield[8].** TTG and greenstone belts constitute approximately 75% of outcropping Rhyacian crust. The remaining 25% corresponds to granitic magmatism, in part Mg-potassic, charnockites, and arc-related dioritic to tonalitic granitoids and gneisses. Late-Rhyacian and Early-Orosirian rocks comprise UHT granulites, charnockites, and high T/P metamorphic rocks partially contemporaneous with the magmatism associated with Cycle 1, marking an increase in silica and incompatible elements in outcropping crust. The following stages are the emplacement of Cycle 2 igneous rocks and the deposition of the continental Roraima Basin, which is intruded by dykes and sills from the Avanavero LIP, correlated to Cycle 3.

partial melting often results from anomalous heat supplied by voluminous mafic underplating, leading to the formation of silicic LIPs[60]. LIPs are well-documented from the Paleoarchean to the present[61] and may have overlapped with subduction episodes in early Earth. Even during times when plate tectonics was inactive or poorly developed, LIP magmatism likely played a critical role in crust-building processes.

While the Orosirian plate tectonic episode is supported by multiple lines of evidence[10], many Paleoproterozoic terranes worldwide lack robust plate tectonic indicators and, instead, consist of rock assemblages that are considered positive evidence of non-plate tectonic regimes[62]. In the Amazon Craton, for example, geological events (illustrated in Fig. 7B) suggest an early stage with transitions from dome-and-keel granite-greenstone terranes—interpreted as products of drip tectonics − to linear belts, which may signify subduction processes in the Archean[51]. Rhyacian events in the Amazon Craton show a close evolutionary relationship with the Birimian Orogen of the West African Craton, where features such as ophiolites and blueschists provide strong evidence of plate tectonic activity[9]. Earlier interpretations of subduction episodes following the Transamazonian−Birimian Orogeny were based on arguments such as juvenile Nd signatures in granitoids and gneiss[26], calc-alkaline geochemical signatures in tonalites and monzogranites[63], "porphyry-like" gold deposits[64], and regional zonation of crystallization/metamorphism ages coupled with Nd model ages[16]. Nevertheless, many of these features are also found in Phanerozoic silicic LIP provinces[60,65,66]. In contrast, diagnostic features such as large-scale radial dyke swarms and short-lived voluminous

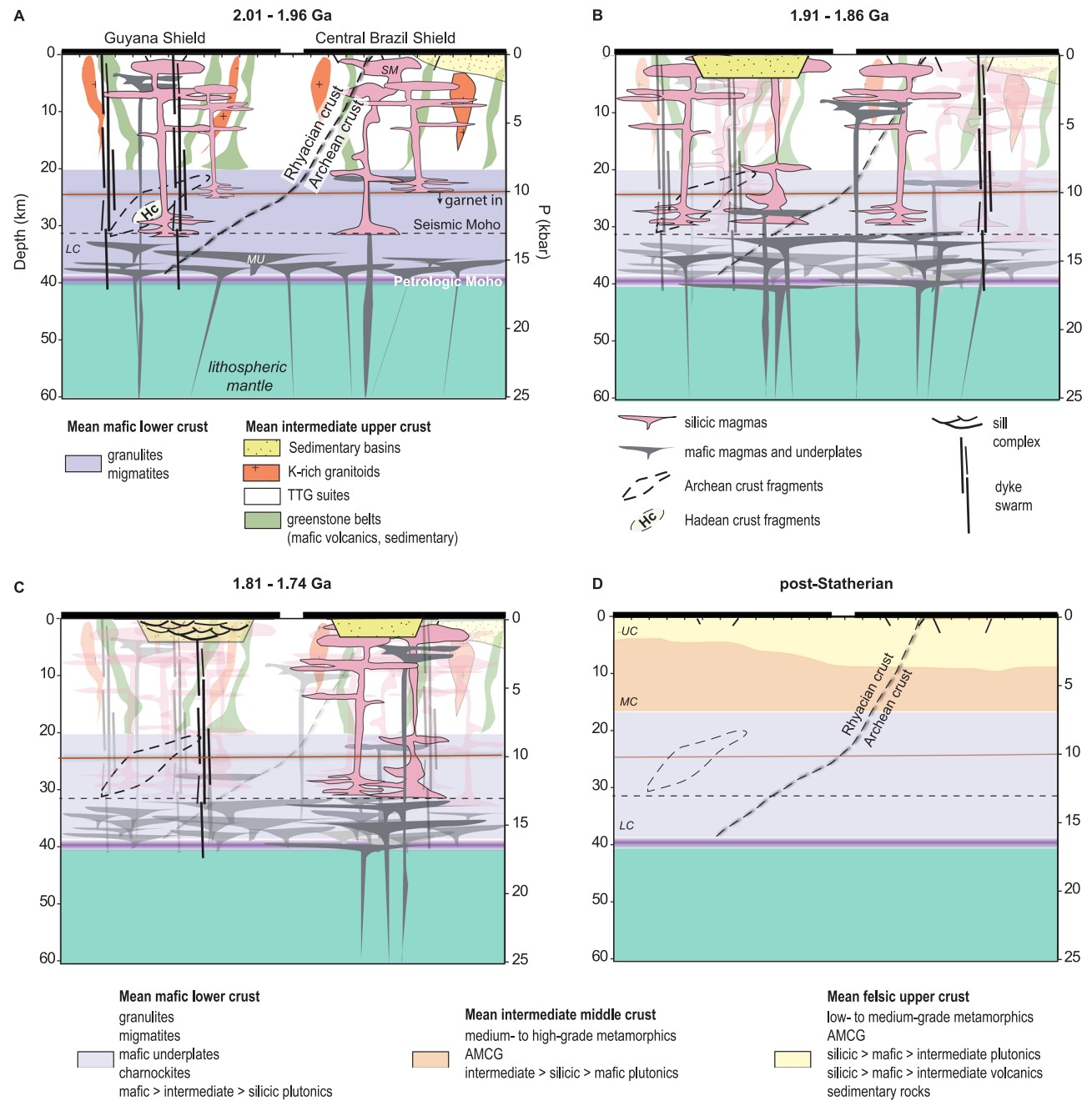

**Fig. 6 | Lithosphere scheme depicting the location of the Guyana and Central Brazil shields across a hypothetical north-south cross-section approximately at 60°W.** The depth of the petrologic Moho around 38 km was calculated using the chemical mohometry model[36]. Mafic underplates (MU) emplaced at the lower crust (LC) were sources of heat for the partial melting of garnet-bearing lower crust material, creating trans-crustal plumbing systems from the lower to the middle (MC) and upper crust (UC). Silicic magma (SM) chambers equilibrating at a mean depth of 5 km to 7 km were the sources of large-scale and province-wide volcanism and emplacement of subvolcanic bodies. **A** Cycle 1 - first LIP outburst. In the Guyana Shield, the basement comprised mainly Rhyacian granite–greenstone and arc associations, truncated Archean fragments, and rare Hadean crust. In the Central Brazil Shield, magmatism developed over an Archean granite–greenstone basement with subordinate Rhyacian arc crust. **B** Cycle 2 - characterized by Rhyacian crustal sources, subsidence, and development of the Roraima Basin in the Guyana Shield, while in the Central Brazil Shield magmatism derived largely from Archean sources. **C** Cycle 3 - in the Guyana Shield, marked by emplacement of basaltic dyke swarms and sill complexes intruding the Roraima Basin (Avanavero LIP). In the Central Brazil Shield, silicic magmatism predominated, with LIP activity preceding intracontinental rifting. **D** Post-LIP crustal architecture - pre-LIP crustal associations (granite–greenstone and arcs) were replaced by a layered crust: (i) mafic lower crust with granulites, charnockites, and underplates; (ii) intermediate crust of medium- to high-grade metamorphic rocks intruded by AMCG complexes; and (iii) an upper, less dense crust of silicic volcanic–plutonic rocks and sedimentary basins.

bimodal igneous suites emplaced across vast regions are not typical of arc-related settings[67].

Geological evidence indicated that the sequence of events associated with and following LIP magmatism included the development of high-T/P (including ultrahigh-temperature, UHT) gneiss belts, the emplacement of Anorthosite–Mangerite–Charnockite–Granite (AMCG) suites, and gabbro–charnockite complexes, with no record of passive margins. These features provided positive evidence for more stagnant tectonic regimes, such as single-lid tectonics[70]. UHT metamorphism and granulite belt formation were likely driven by heat

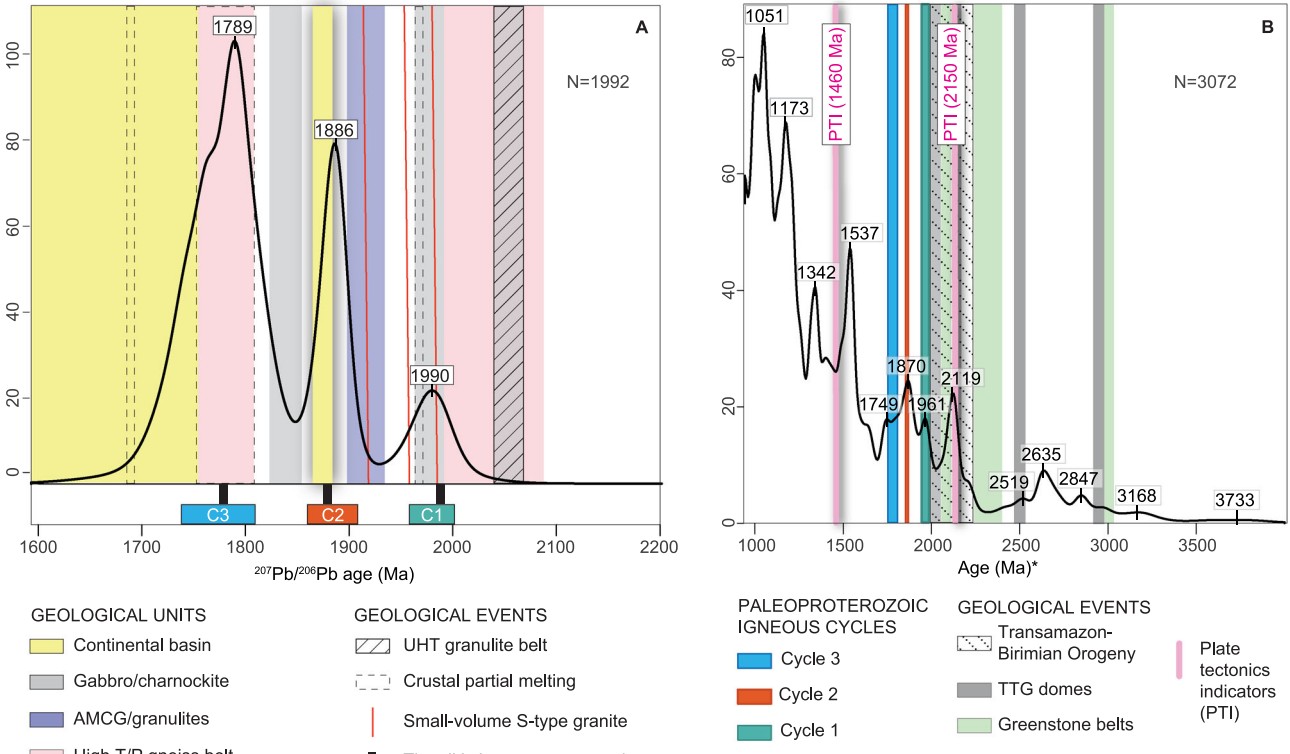

**Fig. 7 | Timeframe of geological events within the Amazon Craton and during LIP magmatism. A** KDE plot of igneous zircons from samples of the volcanic and plutonic units assigned to LIP magmatism and contemporaneous events within the Amazon Craton. During the emplacement of the LIPs, high-grade metamorphism (high T/P), crustal partial melting, emplacement of AMCG and gabbro suites and ultrahigh-temperature metamorphism in granulite terranes took place. S-type granitoids are only found in small volumes during Cycle 1 and preceding Cycle 2. Continental basins developed during Cycles 2 and 3. **B** KDE plot of detrital zircon ages of Amazon Craton sedimentary units and riverbeds from 4000 Ma to 900 Ma, indicating the major peaks of data density. *For zircons older than 1500 Ma, apparent $^{207}$Pb/$^{206}$Pb ages were used, whereas for zircons younger than 1500 Ma,

apparent $^{238}$U/$^{206}$Pb ages were used. The main events include the development of greenstone belts and associated TTG domes during the Archean in the Central Brazil Shield and during the Rhyacian in the Guyana Shield. The Transamazon-Birimian Orogeny is bracketed at 2200–2000 Ma. Cycles of LIP magmatism are assigned and closely compatible with peaks of zircon production at ~ 1960 Ma, ~ 1870 Ma and ~ 1750 Ma. Plate tectonic indicators include two ophiolites: the 2190 Ma Mako ophiolite present in the Birimian Orogen (West African Craton) and the Trincheira ophiolite, with approximately 1460 Ma in the Sunsás Orogen (Amazon Craton). Data for the reproduction of this figure is in Supplementary Data 9.

supplied through basaltic underplating[68]. Collectively, such features reflected elevated thermal regimes[62] and represented unique phenomena restricted to specific intervals in Earth's geological history.

After 1740 Ma, the Mesoproterozoic crustal evolution of the Amazon Craton was dominated by bursts of juvenile magmatism, including gabbroic and AMCG suites, which exhibited minimal contributions from pre-existing crust. This magmatic activity continued from 1600 Ma to 1500 Ma[22]. Plate tectonic indicators, such as ophiolites, appear at 1460 Ma[24], suggesting that subduction processes became active after this period. Prior to this, from the Orosirian to the Statherian, no clear evidence of subduction events exists within the Amazon Craton following the Rhyacian Transamazon-Eburnean-Birimian orogenies. Along with negative evidence of plate tectonics, positive evidence of a more stagnant tectonic regime points out that crustal growth during this interval was primarily driven by large igneous province (LIP) events. A potential exception is the proposed Statherian arc magmatism in the Rio Negro Province[16,69], which lacks definitive plate tectonic indicators and lacks positive evidence for the absence of plate tectonics. This highlights the critical role of intraplate magmatism as a key driver of continental crust differentiation. It functioned as a mechanism to generate crustal strength and stability, facilitating the preservation of large, undeformed, and unmetamorphosed cratonic regions.

## Methods

### U/Pb and trace element measurements

Instrumentation - Photon Machines Excite Excimer 193 nm laser ablation unit coupled to a Nu Instruments, Nu Plasma P3D multicollector inductively coupled plasma-mass spectrometer (for detailed methodology see refs. 70,71).

Operating conditions - samples were analyzed for 20 seconds using a fluence of 1.5 J/cm², a frequency of 4 Hz, and spot size of 20 μm diameter, resulting in crater depths of ~7 μm. Utilizing a standard-sample bracketing technique, analyses of reference materials with known isotopic compositions were measured before and after each set of seven unknown analyses.

Data reduction protocols and secondary zircon standards - corrections for baseline, instrumental drift, mass bias, down-hole fractionation and age and trace element concentration calculations were carried out using Iolite v. 4.1[72]. '91500' zircon (1065.4 ± 0.3 Ma $^{207}$Pb/$^{206}$Pb ID-TIMS age and 1062.4 ± 0.4 Ma $^{206}$Pb/$^{238}$U ID-TIMS age[73]) served as the primary reference zircon to monitor and correct for mass bias as well as Pb/U down-hole fractionation and to calibrate trace element concentration data, while 'GJ-1' zircon (608.5 ± 0.4 Ma $^{207}$Pb/$^{206}$Pb and 601.7 ± 1.3 Ma $^{206}$Pb/$^{238}$U ID-TIMS ages[74]), OG1 ($^{206}$Pb/$^{238}$U ages of 3440.7 ± 3.2 Ma and 3463.3 ± 3.6 Ma respectively[75]), 'Plešovice' (337.13 ± 0.37 Ma $^{206}$Pb/$^{238}$U ID-TIMS age, Temora-2 (416.75 ± 0.24 Ma $^{206}$Pb/$^{238}$U ID-TIMS age[76]) were treated as

an unknowns to assess accuracy and precision. The following U-Pb concordia dates were obtained for secondary reference zircons: 91500: 1064 ± 2 Ma, MSWD = 1.7 ($n$ = 47); GJ-1: 603 ± 3 Ma, MSWD = 0.6 ($n$ = 7); Temora-2: 420 ± 2 Ma, MSWD = 1.7 ($n$ = 7); Plešovice: 335 ± 1 Ma, MSWD = 0.65 ($n$ = 7). An age of 3466 ± 6 Ma, MSWD = 0.06 ($n$ = 7) was obtained from the $^{207}Pb/^{206}Pb$ weighted mean for OG1.

The kernel density estimation and histogram of zircon U–Pb ages, Concordia, and weighted mean date plots were calculated in IsoplotR[77] using the $^{238}U$ and $^{235}U$ decay constants of ref. [78]. All uncertainties are quoted at the 95% confidence or 2 s level and include contributions from the external reproducibility of the primary reference material for the $^{207}Pb/^{206}Pb$ and $^{206}Pb/^{238}U$ ratios; results are summarized in the Supplementary Data 5. Based on the long-term reproducibility of multiple secondary reference zircons, trace element concentrations are accurate to 5% (2σ)[71].

### Lu-Hf measurements

Hf isotopes in zircon were measured in situ by LA-MC-ICP-MS at UCSB. Methods used in this study are similar to those used by ref. [79].

Operating conditions - ablation spots were placed over the original U–Pb analysis spots to obtain Hf compositions that correspond to the measured U–Pb date of the zircon. A laser spot size of 50 μm, a fluence of 1.5 J/cm², and a pulse rate of 15 Hz were used to ablate samples for 30 s, with a 45 s delay between analyses to allow washout. Masses 171–180 inclusive were measured on 10 Faraday cups at 1 a.m.u. spacing.

Data reduction protocols and secondary zircon standards - data were reduced using Iolite v 4.1[72]. Natural ratios of $^{176}Yb/^{173}Yb$ = 0.796218[80] and $^{176}Lu/^{175}Lu$ = 0.02656[81] were used to subtract isobaric interferences of $^{176}Yb$ and $^{176}Lu$ on $^{176}Hf$. The Yb mass bias factor was calculated using a natural $^{173}Yb/^{171}Yb$ ratio of 1.132685[80] and was used to correct for both Yb and Lu mass bias. A natural $^{179}Hf/^{177}Hf$ ratio of 0.7325[82,83] was used to calculate the Hf mass bias factor. Interference and mass-bias corrected $^{176}Hf/^{177}Hf$ ratios were further normalized to 'Plešovice' reference zircon (assuming a $^{176}Hf/^{177}Hf$ value of 0.282482 ± 13[76]). To assess accuracy and precision, natural ('91500', 'GJ-1', 'Plešovice', 'Temora-2', and 'Mud Tank') reference zircons were run concurrently and treated as unknowns. The weighted mean corrected $^{176}Hf/^{177}Hf$ values (±2 SD) obtained for the secondary reference materials are: 0.282312 ± 22, MSWD = 1.3 ($n$ = 12) for 91500, 0.282006 ± 23, MSWD =1.7 ($n$ = 9) for GJ-1, 0.282483 ± 13, MSWD = 0.36 ($n$ = 13) for Plešovice. 0.282671 ± 24, MSWD = 1.8 ($n$ = 13) for Temora-2 and 0.282500 ± 18, MSWD = 2 ($n$ = 11) for Mudtank in good agreement with preferred values.

Reference parameters - epsilon Hf at time t for each analysis was calculated from the inferred crystallization age for each sample (except for inherited grains, from which spot ages were used), using a $^{176}Lu$ decay constant of 1.867 × 10-11 a-1[84] and CHUR values of[42]. Reference materials used for monitoring trace-element data quality were NIST612[85] and BHVO[86] glasses.

### Data availability

All data generated or analyzed during this study are included in this published article (and its supplementary information files).

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

## Acknowledgements

We thank the National Council for Scientific and Technological Development – CNPq (project 404063/2021-7) received by M.S.S. and Fundação de Amparo à Pesquisa do Estado do Rio Grande do Sul – FAPERGS for a post-doctoral grant n° 150171/2023-4 received by M.S.S. and project funding n° 23/2551-0000139-0 received by C.A.S.

## Author contributions

M.S.S and M.L.V. conceived the project. M.S.S., M.L.V., L.M.M.R. and T.A.M. acquired samples. M.S.S., A.R.C.K-C. and J.M.C. performed the analysis. The paper was written by M.S.S. with contributions and reviews from A.R.C.K-C., M.L.V., C.A.S., L.M.M.R., J.M.C., and T.A.A.M.

## Competing interests

The authors declare no competing interests.
