## [Transparent Peer review file · Nature Communications]

Large-scale crustal growth driven by LIP magmatism during the Paleoproterozoic

Corresponding Author: Dr Matheus Simões

A version of this paper was originally rejected for publication by Nature Communications, however that decision was reconsidered after appeal by the authors.

Version 0:

Reviewer comments:

Reviewer #1

(Remarks to the Author)

Dear Simões et al.,

Your manuscript "Large-scale crustal growth determined by LIP magmatism during the Paleoproterozoic," examines the role of Large Igneous Provinces (LIPs) in the Paleoproterozoic crustal growth of the Amazonian Craton. The study bases its interpretations and conclusions on a significant set of zircon U-Pb, Lu-Hf, and trace element data from volcanic rocks, taken as representative of three Paleoproterozoic silicic LIPs, along with compilations from the literature. However, substantial revisions are required to improve clarity, methodological rigor, and the interpretation of results before the paper can be considered for publication.

I have subdivided my review into two parts: analytical data and the manuscript. The detailed review of both parts can be found, respectively, in the attached word files "Revision_SuppMaterial" and "Revision_Manuscript."

When reviewing a manuscript with isotopic data, my first step is always to check the reference materials and attempt to reproduce the figures and age calculations. The isotopic data for reference materials are in excellent agreement with accepted values. However, I encountered several discrepancies that need to be addressed to ensure reproducibility and clarity of the results of the studied.

My main concern is that the plots and age calculations presented for the studied samples are not reproducible. Although the calculated ages are consistent within uncertainty, the plots differ. The possible reasons for this are explained in more detail in the attached word file "Revision_SuppMaterial.". You must ensure that the data presented can be reproduced by the reviewers (and future readers). The supplementary data tables need to be updated to ensure clarity, reproducibility, and proper interpretation of the data.

Regarding the manuscript, nearly all sections, including some figures, need substantial revisions. Please refer to the detailed review in the attached word file "Revision_Manuscript."

To summarize, the revised version should:

- i) ensure full reproducibility of the figures presented in this study by completing the reported data of the studied samples and providing the data resulting from literature compilation;
- ii) provide CL images of representative zircons of each samples, with location of the analysed spots.
- iii) improve the visualization of some figures and, importantly, include geological maps;
- iv) most importantly, explore whether the role of the LIPs in the crustal growth of the Amazon Craton necessarily contradicts the currently accepted accretionary orogenic model. Rather than dismissing one process in favour of the other, the manuscript could discuss how both may have contributed to the crustal growth of the Amazon Craton.

I believe the suggested modifications in the word files are feasible within the current scope of your manuscript and would significantly strengthen its contribution to the debate on the role of LIPs in the Paleoproterozoic crustal growth of the Amazon Craton.

Given the number of revisions needed to clarify inconsistencies, improve data presentation, and deepen the discussion, my recommendation is major revision.

Reviewer #2

(Remarks to the Author)

The manuscript "Large-scale crustal growth determined by LIP magmatism during the Paleoproterozoic" submitted for publication in Nature Communications, was evaluated. The manuscript presents a critical and original discussion about the evolution of Orosirian LIPs in Amazonian craton, focusing on its importance to crustal growth. The manuscript is well presented, structured, and concise. In general, it is well written and illustrated, and based in a current literature. The Abstract summarizes the general questions and conclusions of the manuscript. The manuscript presents new geochemistry, isotopic (U-Pb and Lu-Hf) and Ti-thermometry data for zircon, but discussion about tectonic aspects are almost absent. All along the manuscript, the authors use the term "Amazon craton". The correct expression is "Amazonian craton", as it was initially proposed by Almeida et al. (1981) to define the same region and setting.

The main constraint of the manuscript is the lack of tectonic evidence, among others, compatible with SLIP setting for the study regions. Sometimes, only geochemistry data lead to dubious conclusions. According to the chemical composition of the rocks (mainly calc-alkaline to shoshonitic, high-K, I-type to A-type), it is possible to accept these rocks as occurring in pre- to post-collisional, as well as in an intraplate (fissural) environment. In fact, several authors argue magmatic arc setting for the rocks included in the three cycles proposed in the manuscript. It is the case of Fraga et al. (2020, 2024) for the Orocaina Igneous Belt (cycle 1); Almeida et al. (2001), Vasquez et al. (2002), Santos et al. (2004), Juliani et al. (2013), Fernandes and Juliani (2019) for the Tapajós and Iriri-Xingu domains (cycles 1 and 2), and Cordani and Teixeira (2007) for Juruena region (cycle 3). In the manuscript, the SLIP-related setting is accepted for all regions without discussion. Arc magmatic and intraplate events may simultaneously or sequentially happen, since well-arranged in the space and time. For instance, in the Tapajós domain, where rocks from cycles 1 and 2 occur, and also magmatic arcs are argued, it is necessary reconstruct the tectonic history, fit both cycles and prove if they are true SLIPs. In summary, I am not against the SLIP hypothesis, but it is necessary show evidence in a comprehensive geological setting. They could be shown in tables or as Supplementary Material.

Another discussion needing to be enlarged concerns the extensions and the representations of the three cycles in Figure 1. In the Tapajós and Iriri-Xingu domains (southeastern part of the craton), rocks from both cycles 1 and 2 occur. In my point of view, it is not possible separate two different domains, as it is shown in Figure 1, because in some areas there are both cycles, side-to-side, overlaying or cross-cutting one over the other. The same is true to Peixoto de Azevedo domain, surrounded by rocks from cycle 3, where rocks from cycle 2 is also present. Additionally, a 1.88 Ga granite magmatism is spread in the Carajas domain, extreme southeastern part of the craton. This magmatism has been correlated to Uatuma SLIP, present in the adjacent Iriri-Xingu and Tapajós domains. In line 331, a rock from cycle 2 occurring in Tucuma area (Carajas Province) is exemplified, but it is not shown in figure 1. I suggest including as Supplementary Material a map presenting all sample locations, which data were used in the manuscript.

Version 1:

Reviewer comments:

Reviewer #2

(Remarks to the Author)

The reviewed manuscript "Large-scale crustal growth determined by LIP magmatism during the Paleoproterozoic", was evaluated. The manuscript has been substantially improved, especially with the inclusion of new maps and supplementary material. Now, it is clearer, precise and denser. However, I have some additional comments, hoping the manuscript to be better.

- I suggest including in the title "Amazonian Craton", since the object of the manuscript is exclusive over this region.
- Indeed, in my first review, in according to Stratigraphic code, I suggested replacing "Amazon Craton" by "Amazonian Craton", term originally proposed by Almeida et al. (1981) to define the same region and setting. This point was not commented by the authors.
- The new "Arc vs LIP" table summarizes a comprehensive characteristic set helping the discussion about the tectonic setting. However, it is possible to see that they are not conclusive, in special, concerning poorly known regions, as we have in the Amazonia Craton. For instance, considering this constraint, the debate Arc vs LIP is not a crucial obstacle to advance the discussion. The main aspects to support the manuscript approach are the huge magma volumes with age, isotope and chemistry compositions nearly well determined.
- Concerning this aspect, I believe that the statement "...since no plate tectonic indicators have been identified" (line 71) is too strong to the actual level of geological knowledge of the Amazonian Craton.
- In Fig. 1, why were the Paraguá and Rio Apa domains excluded from the Amazonian Craton? Most authors accept Paraguá domain as allochthonous block, but it is not a reason the exclude it. For example, the Archean Imataca and Amapá blocks probably are also allochthonous, but were considered as part of the craton.
- In relation to the geochronological provinces, it is important to highlight that not all of them are considered as orogenic belts

(line 132). It is the case of the Carajás domain, as well as the Iriri-Xingu and Erepecuru-Trombetas domains, all of them included in the Central Amazonia Province. Moreover, the NW-trend Iriri-Xingu and Erepecuru-Trombetas domains are almost fully composed by rocks from cycles 1 and 2.

- Still concerning the geochronological provinces, not all of them are NW-trending orientation (see Response for Reviewer 2). It is true for provinces younger than 2.0 Ga, but it is not the case of the Maroni-Itacaiunas Province (Guiana Shield).
- I disagree that the "...the Orosirian-Statherian orogenic belts do not match the orientation of N-S and E-W Orosirian-Statherian igneous belt" (lines 132 and 133). Cycle 1 occurs in the E-W Maroni-Itacaiunas Province, as well as in the NW Central Amazonia Province, this last one completely composed of rocks from the cycles 1 and 2. Cycle 3 crops out in E-W southern part of the Rio Negro-Juruena Province. However, it doesn't matter if the igneous and orogenic belts match or not. The SLIP magmas could utilize previous regional structures to penetrate the upper crust.
- In regard to lines 533-535, there are several authors reporting evidence of plate tectonics in the Rio Negro Province (e.g. Almeida et al. 2022 (<https://doi.org/10.1080/00206814.2021.2025158>), and reference therein).
- Reference 340 (line 252) and fig. 2d (line 269) do not appear in the manuscript.

Reviewer #3

(Remarks to the Author)

Simões et al., 2025. Nature Comm.

General comments

The manuscript convincingly argues that LIPs were major contributors to Paleoproterozoic crustal growth. However, has this not been shown previously by for example the work of reference 12–14?

The idea of mantle cyclicity at ~95 Myr intervals aligns well with emerging models of deep Earth dynamics. (Just be honest with the ages here)

It places the Amazon Craton in a global context through correlations with West Africa, Baltica, and others—strengthening the case for a broader LIP-linked geodynamic system.

I am a bit puzzled why the authors only use, and base their discussion on, Nd model ages, when their own data is zircon Hf data and model ages can be calculated from these data. Is there a reason for this? The authors claim to have calculated two stage model ages (although undefined) but do not present them or seem to use them.

The abstract could benefit from more clarity by directly stating the key finding: that LIPs contributed significantly to crustal growth via deep-crustal partial melting.

The discussion could be better segmented to explicitly highlight contrasts between arc and LIP-related crust formation models.

While the authors argue for intraplate LIP-driven crust formation, the paper could further address counterarguments from proponents of arc-dominated growth models.

The interpretation of zircon trace elements as direct indicators of source pressure and temperature, though accepted, would benefit from more explicit discussion of uncertainties and assumptions.

Comments on data

I cannot recreate the ϵ_{Hf} values for the data.

The authors do not define what values for, for example CHUR was used. (Bouvier et al., 2008)?

Using the same decay constant and bouvier CHUR values I get quite substantially different ϵ_{Hf} values than the authors. This is VERY concerning! Even using other (common) CHUR values I doubt I would get the same values as the authors.

In several places (for example row 28 (VR05-1)) there are Hf/Hfi data and a calculated ϵ_{Hf} value. But for that analyses (and several more) there are no raw data in columns CF–CO. The table is generally very confusing and difficult to decipher. ϵ_{Hf} values are calculated for zircon with no data and vice versa.

Even though not stated the ϵ_{Hf} values seem to be calculated at the individual zircon dates (based on spot locations on figure 4). This has been shown multiple times that this is inaccurate. See for example Whitehouse and Kemp (2010, Geol. Soc. Spec. Pub. London).

ϵ_{Hf} values should always be calculated at the interpreted age of igneous crystallisation.

The "preferred age" for several samples according to the summary table on tab Sample identification state that they prefer and use the "concordia weighted mean age". What is that? Either it's a concordia age or a weighted mean age. And if a weighted mean age, then it needs to be defined as for example $^{207}\text{Pb}/^{206}\text{Pb}$ or similar. If this is a weighted mean of individual Concordia dates (which is unconventional, but would be ok) this needs to be explained, and the individual concordia dates should be added to the data column in the supplementary table.

The only form of explanation is found on lines 201–202: "Ages are reported here as the maximum likelihood estimate with its analytical uncertainty/analytical uncertainty with dispersion.

Furthermore: 6/7 weighted mean in Supp. table 2. Tab "Sample identification" should be 7/6 I assume.

OGC $\epsilon_{\text{Hf}} = +0.6 \pm 0.2 \text{ 2s}$ (Kemp et al., 2017), in this study the results come out to: $\sim +2.7 - 2.5$ units too high. Is this data correct?

Some reference zircon have data for Hf in column T–W, but some only in row CF and forward. What is the difference here? OGC have no calculated ϵ_{Hf} values, but there is raw data. It's all very confusing, and the fact that these reference zircon are giving the incorrect values combined with the unreadable table makes me worried.

(However, the data in row CF in the "data" tab does not seem to be the measured values, as this doesn't make sense for any of the reference zircon) This needs to be marked more clearly so that one can check the data quality of the reference material.

"To assess accuracy and precision, natural (91500', 'Mud Tank', and GJ-1) and synthetic (MunZirc-1, -3, and -4) reference zircons were run concurrently and treated as unknowns."

This data is nowhere to be found! At least not for Mud Tank and MUN zircon.

What lab was used for these isotope analyses? The machines are listed, but not what lab.

How was the primary standard values for zircon Hf data checked? Against JMC-475? Not reported. Also, the $178\text{Hf}/177\text{Hf}$ is higher than usual in these analyses. A mean value of all analyses is 1.4677 when the reported value is 1.4772 for, for example JMC-475. All individual analyses have a $178\text{Hf}/177\text{Hf} >$ than 1.4772. Please comment on this.

What was used in the Yb correction? Among the data presented in the table there is no zircon with high Yb which would be usable for Yb corrections. OGC would be good. Mud Tank would be good. MUN would be ok. But none of this data is presented.

Individual line comments

Line 14: The rise of plate tectonics in the Paleoproterozoic. A LOT of studies would argue against this, but here its stated as a truth. This needs to be addressed.

Line 65: There are basically no 2.2 Ga rock in Fennoscandia. To find 2.2 Ga old bedrock in the region, you only find it in the eastern and northern parts, specifically within the Karelian Craton, in areas of Finland and Russian Karelia (In this paper the Kola-Karelian) where Paleoproterozoic supracrustal sequences, often associated with early rifting events, are preserved.

Lines 74–75: Why 95 Myrs.? The timespan between these magmatic episodes are 100, 100 and 110 Ma.

Line 158: Is negative Nb-Ta anomalies really likely to be present in LIPs? You need either significant crustal contamination during magma ascent and emplacement, or fractionation of Ti-bearing minerals within the magmatic system. I would rephrase this sentence.

Line 209–: Results are written in past tense (yielded) but these analyses still yield. Hence, this should be in present tense (yield).

Line 231: Add these ratios as columns in supplementary table to easily check. A mean Th/U of 1 is very high. Also, the word "ratios" is redundant as the Th/U and Yb/Sm are ratios. Hence, this reads "ratio ratio".

Lines 291–295: Statement requires references.

Line 300: What DM was used? What CHUR values were used?

Lines 303-304: "Primary sub-populations for Cycle 1, identified in KDE plots, occur at -5.4 and +2.7." 1. What does this mean? 2. Occur?

Line 305: clustering at -0.89 is very specific, please rephrase.

Line 306: haspredominantly (missing space)

Line 311: Figure 3? Should be figure 4, right?

Line 315: What is clustering here at the specific value of +0.68? All data? Some of the data? It doesn't look like a tight cluster. If this is a mean value I assume the MSWD is rather high if all data is included. If all data is not included, then this needs to be stated.

Oh, now I saw the curve in the figure. So this is a peak, not a cluster... Please explain this more carefully.

Fig. 4: What DM was used in the model age calculations for Nd data?

Line 444: These are not only zircon U-Pb dates. Baddeleyite have been used to date at least a few of these, which is apparent even from the titles of the references.

Line 446: Well. The time difference between the direct ages of the dykes are 100–110 Myrs. The difference between the zircon peaks, which I assume have an uncertainty that overlaps the baddeleyite/zircon ages of the dykes might be ca. 95 Myrs. But these are peaks, not direct ages. I think the authors needs to be more up front about this. Ca 100 Ma is still a good match to your harmonic mantle cycles, but it would be a stronger argument if based on the direct dates presenting the “actual” age gaps. However, the age interval presented and forwarded in the cited reference [58] is NOT 94 Myrs but 90 Myrs. So, now the difference is going from 95–94 to from 100 (or 110) Myrs to 90, which is not as strong of a match.

Line 452: This is not ref [58] but rather the work of: S.J. Puetz, K.C. Condie
Time series analysis of mantle cycles Part I: The geologic record in zircons, large igneous provinces and mantle lithosphere
Geosci. Front., 10 (2019), pp. 1305-1326, 10.1016/j.gsf.2019.04.002
These authors claim mantel cyclicity of ca 93 and 187 Myrs.

Line 466: You present data, not evidence. Evidence is a very strong word choice. I would suggest rephrasing.

Lines 480–488: Recent studies have shown, using several different basis of evidence that there is a lot of water in the mantle. Hence, I would strongly argue that subduction is NOT a prerequisite for the generation of silicic magma. See for example papers below.

1. Schmandt, B., & Jacobsen, S. D. (2014). Dehydration melting at the top of the lower mantle. *Science*, 344(6189), 1259-1262.
2. Pearson, D. G., Brenker, F. E., Nestola, F., McNeill, J., Nasdala, L., Hutchison, B. J., ... & Stachel, T. (2014). Hydrous mantle transition zone indicated by ringwoodite from Brazilian diamond. *Nature*, 507(7491), 221-224.
3. Wang, W., Schmandt, B., & Hu, J. (2025, forthcoming or recent). Water Stored in the Mantle for Millions of Years May Be Linked to Continental Volcanism.
4. Bolfan-Casanova, N. (2018). Water in the Earth's mantle. *Mineralogical Magazine*, 69(3), 229-257.

In summary, the scientific consensus over the last decade has solidified the understanding that the Earth's mantle, particularly the transition zone, is a vast reservoir of water, likely holding several times the volume of our surface oceans. This understanding is built on a multidisciplinary approach combining seismology, high-pressure mineral physics, and geochemistry.

Figure. 1A. The colour difference between the eclogite and ophiolites are too small. I would recommend changing these (or one of these) colours.

Figure. 2. I cannot find dyke swarm C2 in the figure.

Figure 3C. Are the error bars 1 or 2 sigma?

Figure 4. The authors mention second stage TDM ages. These are: 1. Not defined by $^{176}\text{Lu}/^{177}\text{Hf}$ for the second stage. 2. They are not presented anywhere.

Figure 5. Why are charnokites and 2.11–2.08 granitic magmatism denoted using magma blobs instead of squares like other magmatic events/intrusions, like for example the 2.0–1.93 Grnaultite, charnokite, high T/P gneisses?

Figure 6. The evolution of the figures from A–D are not discussed. Please expand the figure caption to guide the reader what is happening in the transition between these stages.

Figure 7. The colour difference between the greenstone belts and cycle 1 is too small in (B).

Version 2:

Reviewer comments:

Reviewer #3

(Remarks to the Author)

I am happy with the new, updated data table and how the data treatment has been updated.
As all my major issues have been dealt with I am now happy with this manuscript proceeding to publication.

Response to Referees

**Large-scale crustal growth determined by LIP magmatism during the
Paleoproterozoic**

Matheus S. Simões, Carlos A. Sommer, Marcelo L. Vasquez, Lucas M.M. Rossetti, John M. Cottle, Túlio A. Mendes, Andrew R.C. Kylander-Clark

Response to reviewer 1 – Part I

Introduction

Lines 27-30: “The period from the Late-Rhyacian to the 27 Statherian includes anachronistic events of rift-drift, accretionary and collisional orogenies in most Paleoproterozoic belts, with episodes of silicic additions to the continental crust. An exception to these belts is the Late-Rhyacian to Statherian record of the Amazon Craton in South America.”

Why is the Amazon Craton record an exception? As you mention further down, in the geological context, the evolution of the Amazon Craton has long been interpreted as the result of southwestward continental growth, characterized by the continuous migration of tectonic and magmatic processes from an ancient Archean core in the east. This process is thought to have occurred through successive magmatic arcs accreted along the southwestern margin of the craton or the development of a long-lived accretionary belt (e.g., Tassinari and Macambira, 1999; Santos et al., 2000; Cordani and Teixeira, 2007; Scandolara et al., 2017; Trevisan et al., 2021; Motta et al., 2022), in line with the broader concept of the Great Proterozoic Accretionary Orogen (Condie, 2013).

We appreciate this initial comment and understand its relevance. However, we emphasize that the introduction is intended to frame the broader rationale of the manuscript. The question of “why the Amazon Craton is an exception” is addressed progressively throughout the text.

To provide further context, we have included a new supplementary table (Supplementary Table 2) in the revised manuscript that summarizes arguments supporting or challenging both LIP- and subduction-related models. In brief, the first three references cited by the reviewer are regional in scope and primarily rely on age rejuvenation and variations in isotopic ratios from different systems. Recent studies, along with our new data, suggest that such patterns are not universally applicable—exceptions exist, and the association of geochronological provinces with orogenic activity remains a subject of ongoing debate in the context of the Amazon Craton (AC) and should not be treated as an “*a priori* truth”.

For instance, Scandolara et al. (2017) present a detailed study of a portion of the AC proposing an arc-related setting without any direct evidence of subduction. Their interpretation is based on airborne magnetic data from remote, inaccessible areas (interpreted as thrusts without field evidence) and geochemical signatures (high-K calc-alkaline rocks with negative Nb-Ta anomalies), which can also occur in intraplate environments.

In contrast, Barros et al. (2009) and Rizzotto et al. (2019) offer a different interpretation of the same rock units. These works emphasize the bimodal nature of the magmatism based on the geochemistry of igneous rocks, and on the proposal of a metamorphic core complex formed in an extensional tectonic setting.

Regarding the contributions of Trevisan and Motta, we acknowledge that their work offers valuable new data and insightful reinterpretations. Nonetheless, the same geochemical parameters invoked in support of arc settings—high-K calc-alkaline signatures, negative Nb-Ta anomalies, and tectonic discrimination diagrams—do not provide unequivocal evidence for subduction. Moreover, the occurrence of porphyry-like mineralization is typical of Phanerozoic arc environments, but is not diagnostic of an arc setting, as similar mineralization is observed in silicic LIPs such as the Sierra Madre Occidental (Camprubí et al., 2017 - <http://dx.doi.org/10.1016/j.oregeorev.2016.01.006>).

Several of these debates have already been highlighted in prior literature (e.g., Rizzotto et al. works) and the presence of silicic LIPs within the Amazon Craton has been previously proposed based on robust datasets (e.g., Teixeira et al., 2019). Therefore, we maintain that our interpretation of the existing literature, combined with new data presented herein, is scientifically well supported and consistent with ongoing discussions in the field.

Lines 31-33: This is a key area to investigate mechanisms of crustal growth and a potential relationship with subduction since **no plate tectonic indicators have been identified. Instead**, the period from 2.0 Ga Ma to 1.74 Ga Ma in this crustal segment is marked by the emplacement of large igneous provinces with an extensive bimodal igneous association.

This sentence likely needs rephrasing, as many will disagree with its current wording. What do the authors consider to be “plate tectonic indicators”? Arc-related geochemical signatures in Paleo- to Mesoproterozoic magmatic rocks, along with increasingly positive ϵNd and ϵHf values with decreasing U-Pb crystallization ages away from the craton are typical characteristics of accretionary orogens. Additionally, “instead” may not be the most appropriate transition. A more suitable alternative would be: "Between 2.0 and 1.74 Ga, the Amazon Craton also records the emplacement of large igneous provinces with extensive bimodal magmatism."

It is possible that some researchers may disagree with our statement regarding Plate Tectonic Indicators (PTIs) in the Amazon Craton. However, such disagreement does not, in itself, constitute a compelling reason to rephrase the statement. As outlined by Stern (2023), PTIs include features such as ophiolites, eclogites, passive continental margins, tall collisional mountains, paleomagnetic constraints, hydrous fluid-sensitive ore deposits, S-type granites, and deep crustal geophysical imaging.

In contrast, arc-like geochemical signatures and isotopic zonation alone are not definitive evidence of arc-related tectonic settings (e.g., Ewart et al., 1992 - <https://doi.org/10.1017/S0263593300008002>). In our work and several others (e.g., Motta et al., 2022), there is evidence of extensively negative ϵNd and ϵHf values, so the argument on these features is not consensually valid. High-K calc-alkaline rocks, for instance, can also form in intraplate environments, along with A-type magmatism (Bryan, 2007; Ewart et al., 1992; Ferrari et al., 2018). Moreover, isotopic zonation can arise within blocks of continental crust that have been repeatedly intruded by both mafic and silicic magmas. These intrusive events may reset low-temperature isotopic systems or modify the underlying mantle composition, thus shifting ϵNd and TDM model ages over time (Lu et al., 2024 - <https://doi.org/10.1029/2024GL108715>).

I will return to this point later, but I believe a key aspect of the manuscript should focus on whether the presence of LIPs/SLIPs necessarily contradicts the widely accepted accretionary orogeny model for the evolution of the Amazon Craton. Alternatively, could these processes be reconciled? If so, craton growth could be understood as the result of both mechanisms—horizontal growth through accretionary processes and vertical growth associated with the emplacement of LIPs/SLIPs away from the craton margin (subduction zone). As noted by Teixeira et al. (2019), LIPs/SLIPs “appear to accompany the stepwise accretionary crustal growth of Amazonia,” and several other authors have linked them to slab retreat events.

We agree that these processes can be complementary. In our manuscript, several rock units dated between 2200 and 2020 Ma—such as those from Cuiú-Cuiú, Jacareacanga, and the Guyana Shield—which are already interpreted as products of orogenic activity related to the Transamazon Orogeny in the literature, remain with this interpretation. This orogenic episode is further supported by the presence of a plate tectonic indicator (ophiolite) in the West African Craton. Without this phase of orogenesis, the necessary conditions for subsequent silicic LIP magmatism would likely not have been achieved.

Regarding slab retreat events, we respectfully note that clear evidence of subducted slabs—aside from the recognized ophiolites at ~2150 Ma and ~1460 Ma—is lacking. The proposed accretionary models are not currently supported by sufficiently compelling data, and as such, we do not consider there to be a strong scientific basis for adopting these models as the default interpretation.

Geological context of the Amazon Craton

General comment:

This section would greatly benefit from a simplified geological map of the Amazon Craton. It is extremely difficult to follow the text without a figure (and I am rather familiar with the Amazon Craton).

The figure should display: i) the main tectonic provinces of the Amazon Craton, with their ages indicated in the legend; the superposition of the studied SLIPs; iii) the locations of the analysed samples. Additionally, since the "SAMBA" model is referenced, it would be useful to include it as an inset within the geological map.

A new figure (Fig. 1a) was made using the SAMBA model and the geological map of the Amazon Craton (Fig. 1b) with emphasis on the studied magmatic provinces, as suggested.

Furthermore, given that you have a detailed map of the Guiana Shield (current Fig. 4), it would be appropriate to provide a similar one for the Central Brazil Shield, as most of the studied samples come from this area.

A detailed map of the Guyana Shield is not included in the paper solely to indicate sample locations. Rather, it was specifically compiled to highlight Rhyacian dome-and-keel structures, orogeny-related rocks, and large igneous provinces (LIPs). Since the Central Brazil Shield does not display Rhyacian dome-and-keel structures, a detailed representation of this area was deemed unnecessary for the purposes of this manuscript. Nevertheless, geological maps with sample location are now presented as supplementary material (Supplementary Data 1).

Lines 59-53: "The distribution of Archean crust is evidenced as an isotopic provinciality that divides the Amazon Craton into an eastern segment, characterized by Archean Nd depleted mantle model (TDM) ages and the most negative $\epsilon\text{Nd}(t)$ values, and a western segment, where Rhyacian to Orosirian Nd TDM ages and slightly negative to slightly positive $\epsilon\text{Nd}(t)$ values predominate (Fig. 1a, b)."

This figure does not seem to be the most appropriate choice for this section. A Nd TDM age map is not necessary to highlight the Archean crust and the Paleo- to Mesoproterozoic provinces; a geological map is much more effective. To enhance clarity, as mentioned earlier, the SLIPs and sample locations should be plotted on a geological map, rather than on an Nd TDM age or $\epsilon\text{Nd}(t)$ map.

We acknowledge that Nd isotope maps are valuable for illustrating the progressive rejuvenation of the underlying crust and mantle throughout each magmatic cycle. We included geological maps with sample location as supplementary material (Supplementary Data 1).

Comments about Fig. 1:

“The database for contour construction is sourced from 27 and 21.”

i) Reference '27', Condie et al., 2021, does not include any Nd isotopic data. Similarly, '21', Motta et al., 2022, only provides Nd data for the Central-Brazil Shield. Could the authors clarify the source of the remaining data?

The correct reference of Nd isotope data is Puetz and Condie (2021) Volume 10, Issue 4, July 2019, Pages 1305-1326– Multimedia component 3. This is now corrected in the text and references.

ii) If the data is from different sources, did the authors confirm that the TDM ages were all calculated using the same Depleted Mantle model? If not, they should consider recalculating them for consistency.

The dataset from Puetz and Condie (2021) includes a tab detailing the specific models used for isotopic calculations, while the Motta et al. (2022) compilation presents values as originally reported by each author. We agree that applying a uniform model across all datasets would enhance consistency. However, the degree of variation introduced by using different depleted mantle (DM) models does not significantly affect the overall interpretation of the map. The spatial patterns and isotopic variations remain valid, and the data from the Central Brazil Shield show good correlation with those from the Guiana Shield, as illustrated in the figures.

iii) The Nd data used should be provided as supplementary material (figures must be reproducible by the reviewers and future readers). The figure should also include the sample positions so that the reader can assess the sample density and distribution.

The isotope data used for the map is now presented as a new supplementary table (Supplementary Table 1), as suggested.

iv) The software used and the interpolation method applied for generating the data in the figure should be specified.

This is now specified in the figure capture, as suggested.

Lines 75-79: “Nevertheless, the record of plate tectonic indicators in the Amazon Craton comprise a 1460-1440 Ma ophiolite in the Sunsás Belt, that post-dates Paleoproterozoic belts and pre-dates the late Mesoproterozoic events (1.2 – 0.95 Ga). These events are correlated with the Grenville-type orogens related to the collision of Amazonia and Laurentia”.

I’m not sure why the authors included this or what they intend to convey with this text.

Lines 79-82: Further, the proposition of Orosirian-Statherian orogenic belts in the Amazon Craton does not match with some important surface geological relationships, including two Orosirian and one Statherian igneous belts exhibiting N-S and E-W, NW-SE and E-W trends that conflict with the parallel NW-SE orogenic provinces.”

I don’t fully understand the sentence. From the regional geological maps provided by other authors (since this manuscript does not include one), it is very clear that the provinces are generally oriented in a NW-SE direction. This orientation is well-documented and widely accepted in the published literature. If the authors are suggesting a different orientation, they should provide evidence to support this claim.

The igneous belts are now more clearly depicted in Figure 1b. We are not suggesting that the geochronological provinces are not oriented NW–SE. Rather, we are highlighting that “two Orosirian and one Statherian igneous belts exhibit N–S, E–W, and NW–SE to E–W trends that contrast with the parallel NW–SE orientation of the orogenic provinces.” In this context, our point is that these igneous belts do not share the same structural trends as the geochronological provinces.

Orosirian and Statherian igneous belts:

It is essential to provide at least a brief description of the geology of each of the igneous belts. Are there notable differences between them, or are they very similar? As it currently stands, without referring to the literature, all I know is that there are three Paleoproterozoic SLIPs in the Amazon Craton and their respective ages, nothing more. The lithologies they encompass, their relationships with one another, etc, remain unclear.

We enhanced their description in the ‘Orosirian and Statherian Igneous Belts’ section. Detailed lithostratigraphic review is also presented by Teixeira et al. (2019) and we feel it is not necessary to fully repeat it.

You analysed 15 volcanic rocks, mostly ignimbrites, from three SLIPs. While the supplementary material lists the geological unit of each sample, there is no context provided. The reader is left uncertain about the nature of these formations, whether they are associated with sedimentary rocks, how they relate to the calc-alkaline I-type to A-type plutonic rocks, and so on. There is a significant lack of geological context. In addition, there is an absence of field and petrographic photos of the samples. While the field constraints might be well-established, and I don’t doubt that, this information is not conveyed in the manuscript.

A full report with field and petrographic descriptions along with geological maps for each sampled region is now presented as new supplementary material (Supplementary Data 1).

Results

Age of volcanic rocks

Please refer to the comments in the separate Word file regarding the U-Pb data. Additionally, I believe this section would benefit from a figure comparing your data with previously published data from the literature.

Zircon composition, crystallization conditions and source-depth

The calculated Ti-in-zircon crystallization temperatures (which are not presented as a figure in the manuscript or as a supplementary figure) and ΔQFM values should be provided in the supplementary data table. Additionally, is there any correlation between these values and the U-Pb ages and $\epsilon Hf(i)$ values of the analysed zircons? Additional insights may be gained from this.

Calculated temperature and oxygen fugacity data are now presented in the supplementary material (Supplementary Table 3).

Lines 175-178: “To estimate crustal thickness for each studied magmatic cycle we applied the chemical mohometry model, by using a large dataset of geochemical data available from the literature. Data was filtered to exclude major and mobile trace elements sensitive to hydrothermal alteration processes.”

Data should be presented as supplementary material as it must be available to the reviewers to ensure the calculations are correct.

Geochemical data are now presented in supplementary material (Supplementary Table 4) the models are presented in the Supplementary Data 4.

Lines 180-184: KDE plots of the depth estimate proxy using the $^{176}\text{Lu}/^{177}\text{Hf}$ ratio are not presented as a figure in the manuscript or as a supplementary figure. Results could/should be presented in the supplementary data table.

This is now presented in a figure along the manuscript and as supplementary material (Supplementary Table 3).

Comment on Figures 2b and 2c: if I am not mistaken, the fields were created by the users. If this is the case, I believe the data compiled from the literature to construct the fields should be provided as supplementary data to allow the reviewers to verify the data and ensure the accuracy of the figures.

This is now presented in the supplementary material (Supplementary Table 4).

Hf isotopic signatures from Paleoproterozoic to 201 Ma in the Amazon Craton

Lines 201-204: "In addition to previously published datasets including detrital and igneous zircon age and Hf 203 isotopes, we present 172 coupled zircon U-Pb and Lu-Hf analyses from 15 volcanic rock samples representing the three Paleoproterozoic igneous cycles."

Reference 27, Condie et al. 2021, does not include any Hf isotopic data. Compiled Hf data should be provided as supplementary material and references indicated.

This is now included as supplementary material as suggested (Supplementary Table 5).

Lines 216-244: This part of the text focus on interpretation and implications rather than the presentation of the results, therefore, it would be more appropriate to move this content to the discussion section.

This is now in the discussion section, as suggested.

Comments of figure 3:

i) Detrital and igneous zircon Hf data from the literature are currently presented as grey circles. They should be represented with different symbols for clarity. Igneous zircon should be symbolized according to their geological unit or, alternatively, tectonic province to provide more context and facilitate interpretation.

Since igneous zircon represented less than 20% of the literature data and were obtained for few geological units we decided to remove it from this figure.

ii) What exactly do the literature data for Cycle 1, Cycle 2, and Cycle 3 represent? Why are they separated from the general "detrital and igneous (literature)" data? Given that you only analysed volcanic rocks, mainly ignimbrites from each of the SLIPs, how do these volcanic rocks' Hf compositions compare to the plutonic rocks of each SLIP? Are there any notable differences or similarities between them that could provide additional insights into the geochemical characteristics or tectonic settings of these regions?

We added some new sentences in this part, but different isotopic compositions are only found for Cycle 2, and this difference was explored and partially interpreted in the text.

iii) For example, zircon from the three analysed volcanic samples of the Uatamã SLIP (Cycle 2) show a relatively restricted range of $\epsilon\text{Hf}(i)$ values, from -6.7 to +1.7, with an average value of -2.0. However, the "Cycle 2 literature" data shows a major cluster at -15.0. What lithologies does this cluster represent? Are they the same as the more radiogenic cluster or are they different? Why do the Uatamã SLIP rocks reach much more enriched (more negative) $\epsilon\text{Hf}(i)$ values, approaching the evolution trend of the Archean crust as defined by the "detrital and igneous (literature)" zircon? (By the way, what exactly do these Archean "detrital and igneous (literature)" represent? Igneous rocks from Carajás? Archean detrital zircon from younger sequences?)

We added more information about that.

On another note, how do the Hf isotopic data from this study and the literature compilation compare with the Nd data from the literature? You compiled Nd data for the Amazon Craton, as shown in Fig. 1. Why not compare the two datasets? Do they show similar or divergent trends? This could provide further insights into the evolution of the Amazon Craton.

We added an inset with Nd isotope data from the literature in Figure 3 and presented the comparison of the two datasets along the text.

iv) It is unclear what the starting point ages are for the C-F2, TTG, and KREEP, and why those

particular ages were chosen. Please specify the rationale behind these age selections.

This is now explained along the text.

v) The references for the 'previously published zircon ages' used to construct the KDE diagram must be provided. Additionally, the compilation of age data, coordinates, and other relevant information should be included as supplementary material to ensure transparency and allow for reproducibility.

This is now inserted as supplementary material (Supplementary Table 5). 'Previously published zircon ages' are the same used to constrain Lu-Hf data in the same diagram.

Discussion

Changes in structure and composition of the Amazon Craton crust

The discussion of the Hf data should be moved to this section, and should be discussed along with the Nd isotopic data.

It has been moved to discussion section and is discussed along with Nd data, as suggested.

Lines 292-293: “The isotope composition of silicic igneous rocks reveals that the proportion of reworked crust during magmatism was highly dependent on the availability of pre-existing crustal material.”

While this statement might be valid for Cycle 3 rocks, as you explain in lines 296 to 299, it does not explain the differences observed between Cycle 2 and Cycle 1. Although there is no available projection of the Hf-analysed samples on a geological map, the existing projection on the Nd TDM age map suggests that both Cycle 1 and Cycle 2 rocks are spatially associated with Archean crust, whereas Cycle 3 rocks, further west, overlie a region characterized by predominantly Paleoproterozoic Nd TDM ages. If Cycle 1 and Cycle 2 broadly overlap spatially, their availability of pre-existing crustal material and its characteristics should be comparable. However, the isotopic results indicate that Cycle 2 rocks have incorporated significantly more evolved crustal material than Cycle 1.

This is because there are no Hf analyses on Cycle 1 rocks overlying the Archean domains (e.g., Irixi-Xingu domain). Nevertheless, this is demonstrated in the new Nd data inset on the same figure.

Cycle 3 rocks exhibit an $\epsilon_{\text{Hf}(i)}$ range identical to that of Cycle 1. Identical $\epsilon_{\text{Hf}(i)}$ values for two magmatic events of distinct ages suggest either an increase in the juvenile component and/or a reduction in the influence of the Archean crust in the younger cycle. Given the apparent overall correspondence between the Nd and Hf isotopic data (as seen in both Fig. 1 of this study and Fig. 8 of Motta et al., 2022), and also that Cycle 3 rocks are located west of the Cycle 1 and 2 rocks (i.e., further away from the Archean core), the data strongly suggest that Cycle 3 rocks were emplaced in a distinct crustal “structure” (either younger crust or thinned Archean basement) compared to Cycles 1 and 2.

This suggestion is now incorporated into the main text.

Comments on Figure 5: i) There are no KDEs shown on the right side of the figure. ii) The abbreviations MU, MC, UC, and SM are not indicated in the figure.

KDE's are inserted in a revised figure 2 (now figure 3), abbreviations are now inserted within the figure 5 (now figure 6).

Repeated Cycles of Mantle-Derived Magmatism at ~95 Myr Intervals

Comments on Figure 6: i) Fig. 6A and B are exchanged. ii) Sources of data from the literature must be indicated in the figure caption. Compiled data should be made available as supplementary material.

Figures are now with the correct order in the capture and the compilation is inserted as supplementary material (Supplementary Table 5).

Crustal growth by intraplate igneous events during the Paleoproterozoic

Lines 411-416: "Prior to this, from the Orosirian to the Statherian, no clear evidence of subduction events exists within the Amazon Craton following the Rhyacian Transamazonian-Eburnean-Birimian orogenies. Instead, crustal growth during this interval was primarily driven by large igneous province (LIP) events. A potential exception is the proposed Statherian arc magmatism in the Rio Negro Province; however, this interpretation remains unsupported by definitive plate tectonic indicators."

This is a classic case where the phrase "absence of evidence is not evidence of absence" applies. The absence of "definitive plate tectonic indicators" could be due to a true lack of subduction or simply because of an incomplete record (ophiolite sequences and high-pressure rocks are not very common in the Paleoproterozoic).

In the initial version of the manuscript, we included some elements of *positive evidence* for a non-plate tectonic regime, but these were not sufficiently emphasized. In this revised version, we have clarified the distinction between *negative evidence*—which, on its own, corresponds to the type of argument being invoked in your comment—and *positive evidence*, which complements the negative evidence and strengthens our overall interpretation.

The assertion that crustal growth during the Orosirian-Statherian was "primarily driven by LIP events" is a strong claim that requires robust evidence. While the authors correctly highlight the importance of LIPs in crustal growth, they do not sufficiently address the geochemical and isotopic trends that are commonly associated with accretionary orogens, such as the southwestward decrease in crystallization ages and TDM model ages observed in the magmatic rocks of the Paleoproterozoic Ventuari-Tapajós and Rio Negro-Juruena belts, which in addition exhibit arc-related geochemical signatures. This pattern is a hallmark of long-lived accretionary orogens from the Proterozoic to the Phanerozoic (e.g., Bahlburg et al., 2023 and references therein) and should not be dismissed without a compelling alternative explanation.

We acknowledge that this statement should not be taken as an established truth. It is not typical for arc settings to exhibit features such as radial dyke swarms, bimodal magmatism, and the dominance of A-type silicic magmatism. Moreover, decrease in crystallization ages and TDM model ages are not exclusive indicators of arc-related processes; they may also reflect basement features. For example, even in arc settings, spatial isotopic variation may be attributed to different crustal and mantle sources (Kagami et al., 1992 - <https://doi.org/10.1007/BF00310452>).

For instance, during Cycle 3, we observe TDM model ages spanning the Rhyacian and Archean, which suggest reworking of ancient crust and melting of a Rhyacian mantle source. Additionally, the presence of Orosirian TDM model ages may reflect either the melting of previously formed Cycle 2 crust or mantle rejuvenation processes. These observations reinforce the complexity of the tectonic and igneous evolution and caution against attributing such features solely to arc settings.

Instead of positioning LIP magmatism in opposition to accretionary tectonics, the authors should explore how these processes may have interacted. Could the emplacement of LIPs have played a role in modifying an accretionary orogen? Is there geochemical evidence suggesting interplay between plume-derived and arc-related magmatism? These are the types of questions that could provide a more nuanced and comprehensive interpretation of the Amazon Craton's evolution. A more balanced discussion would acknowledge the role of LIPs while evaluating whether they necessarily contradict an accretionary orogenic model. Rather than dismissing one process in favour of the other, the manuscript could explore how both may have contributed to crustal growth of the Amazon Craton.

We fully agree with the importance of seeking balanced interpretations regarding the Amazon Craton (AC). Naturally, accretionary and collisional events played a role in the AC's tectonic evolution, and we do not dismiss their occurrence. As highlighted in the new supplementary table, several regions of the AC with Rhyacian to Statherian ages now present stronger evidence for intraplate magmatism, even in areas that were initially described as magmatic arcs. The historical view of geochronological provinces is a hallmark for understanding crustal evolution of the AC. Orogenic model-based solutions were a solid response to primarily geosyncline theory-based models. Nevertheless, there were some flaws adopted for reaffirming these models in some places. However, some key assumptions used to reinforce these models warrant reconsideration.

One major limitation lies in the interpretation of whole-rock geochemistry:

1. Most silicic rocks are naturally enriched in K_2O and plot within the high-K to shoshonitic fields on petrological diagrams. So, not an argument for magmatic arc setting.
2. These diagrams have often been used to support Andean-type magmatism without incorporating A-, I-, or S-type granitoid classification diagrams and without considering the geochemistry and evolution of coeval mafic magmatism.
3. A significant volume of these high-K silicic rocks is demonstrably A-type in character, as already established in previous studies, and therefore does not require detailed demonstration in this manuscript.
4. The common association of I-type calc-alkaline or tonalite-granodiorite-diorite suites (contemporaneous to A-type magmatism) with arc settings has often been misapplied. The main misconception regards the lack of detailed geological mapping. For example, single plutons in intraplate settings may record multiple intrusive pulses, alternating between syeno- to monzogranites with A-type affinity and granodiorites/tonalites with I-type affinity—as observed, for example, in the Lavras do Sul (Gastal et al. 2015 - 10.1590/23174889201500020004) and São Sepé (Gastal and Ferreira, 2013 - 10.22456/1807-9806.43440) granites of the Neoproterozoic Sul-Riograndense Shield.
5. Furthermore, many igneous suites—coeval with known A-type magmatism—consist of high-K, calc-alkaline I-type granitoids, which are not exclusive to arc settings, and can also be generated by high-grade metamorphism of deeply buried continental crust and subsequent attendant attendant water-fluxed melting of crustal lithologies (Janousek et al., 2023 - 10.1016/j.jog.2022.101960).

A second flawed assumption involves the interpretation of age zonation in zircon U–Pb data and Nd isotopic signatures as indicative of accretion. The first interpretation of Cordani et al., Tassinari and Macambira, Santos et al., Cordani and Teixeira, assume isotopic zonation across the NW-trending geochronological provinces. Nevertheless, no clear age/isotope ratio zonation is observed for Cycle 1, which is dominantly E-W trending in the Guyana Shield. For Cycle 2, there is no crystallization age zonation, and the Nd isotopic signatures are evidently influenced by underlying Archean or Rhyacian–

Siderian crust (Semblano et al., 2016 - 10.11606/issn.2316-9095.v16i3p19-38; Teixeira et al., 2017 - 10.1016/j.jsames.2017.09.017, and many references from works at eastern AC). Cycle 3 is also E-W trending and exhibits strong evidence for extensional, bimodal magmatic activity and coeval ultra-high temperature (UHT) metamorphism (Rizzotto et al., 2019 10.1016/j.jsames.2019.01.003, Benetti et al., 2024 - link). Nd isotopes and inherited zircons points to a reworked Orosirian crust with local Archean substrate. Later deformation and metamorphic overprints correspond to younger tectonic events unrelated to the timing of LIP emplacement (Almeida et al., 2022 - 10.29396/jgsb), as illustrated in Figure 6.

By stating this, we are not invalidating the geochronological province models. On the contrary, they remain highly valuable for identifying the crustal or mantle sources underlying LIP magmatism. Our work does not seek to directly challenge these models.

Response to Referees

**Large-scale crustal growth determined by LIP magmatism during the
Paleoproterozoic**

Matheus S. Simões, Carlos A. Sommer, Marcelo L. Vasquez, Lucas M.M. Rossetti, John M.
Cottle, Túlio A. Mendes, Andrew R.C. Kylander-Clark

Response to reviewer 1 – Part II

1) When reviewing a manuscript with geochronology data, my first step is always to check the reference materials and to attempt reproducing the figures and age calculations. Results from the reference materials are in excellent agreement with the accepted values.

Using Isoplot-R and following the same conditions outlined in the Methods section, I tested the data reported in the supplementary table for the studied samples. While the calculated ages agree within uncertainty, **I cannot reproduce the age results and diagrams presented by the authors: the plots differ**. Below, I provide an example for sample VR05, though **this issue applies to all samples**. As shown, the calculated ages are consistent within uncertainty, but the plots themselves are different.

While this discrepancy does not affect the overall results of the manuscript, because the calculated ages usually overlap within uncertainty, it is important that any reader should be able to reproduce the figures presented by the authors. **This is an issue and should be addressed**.

I am not sure about the cause of these differences. One possibility is that the authors may have plotted the diagrams using the 207/235 and 206/238 ratios, rather than the 238/206 and 207/206 ratios reported in the table, meaning that the results calculated by Iolite are producing slightly inconsistent isotopic ratios (this is a known issue of Iolite). However, this is just a guess, as **the 207/235 ratio is not included in the data table**, and I cannot confirm this. Depending on how the 207/235 ratio was calculated (mean of ratios versus ratio of means), this could explain the discrepancies observed.

Using sample VR05 as an example, we carefully retraced the plotting procedure and identified a possible cause for the age differences. The analytical errors in our data table are reported as 2σ (absolute), so all diagrams were generated using the 'Input errors' setting as '2se (abs)' in IsoplotR.

However, when we plotted VR05 using the '1se (abs)' option (as shown in the figure below), we obtained the same result as you did. This indicates that the discrepancy arises from selecting the incorrect input error setting during plotting. Therefore, we believe that the mismatch in results is due to this choice in the 'Input errors' parameter and we recommend you try to use 2se (abs) as input.

Input format:

Input errors:

Authors should report the U-Pb isotopic data as follows:

$^{206}\text{Pb}/^{204}\text{Pb}$ (or % of common Pb), $^{207}\text{Pb}/^{235}\text{U}$, $^{206}\text{Pb}/^{238}\text{U}$, rho, $^{207}\text{Pb}/^{206}\text{Pb}$, $^{238}\text{U}/^{206}\text{Pb}$, rho, $^{232}\text{Th}/^{238}\text{U}$.

The complete dataset, arranged as requested, is now included in the revised supplementary

material (Supplementary Table 3).

Plot for sample VR05 reported in supplementary table:

Plot for the same sample VR05 with data from columns B to F:

(Note that in the plot above, I did not reject the youngest and oldest analyses. To obtain an acceptable MSWD I would have had to reject more than 2.)

2) Furthermore:

- Please indicate the type of concordance used (e.g., % difference between the $^{206}\text{Pb}/^{238}\text{U}$ and $^{207}\text{Pb}/^{206}\text{Pb}$ ages, concordia distance, etc.). The authors appear to have used concordia distance (or a similar approach), but this should be explicitly stated.

This is now stated on the Supplementary Table 3.

3) When examining the data for each individual sample, many of the rejected analyses could be interpreted as either significantly older and discordant grains (xenocrysts or inherited cores) or grains affected by common Pb. However, common Pb estimation is not provided in the data table, nor are CL images that could clarify whether these analyses correspond to inherited cores. **CL images of representative zircons of each sample should be provided.**

Zircon CL images are now inserted as supplementary material (Supplementary Data 3).

- Building on the above, when plotting the data for all samples, the common Pb hypothesis seems unlikely, as most data align along discordia lines toward geologically meaningful ages (those younger than 2.2 Ga). In contrast, analyses older than 2.2 Ga are more scattered and usually highly discordant (except for PB07A_8: 2852 Ma), making their $^{207}\text{Pb}/^{206}\text{Pb}$ ages questionable.

4) A potential approach for the treatment of U-Pb data could be considered. As noted by Vermeesch (2021, *Geoscience Frontiers* 12, 843–850), “volcanic rocks often exhibit positively skewed age distributions with a short tail of syn-eruptive U–Pb ages and a long tail of pre-eruptive or xenocrystic ages.”. In such cases, the maximum likelihood age model (minimum age on radialplot of Isoplot-R) might be a good alternative for determining the crystallization age, as it statistically rejects possible xenocrystic ages. This automated method is likely more accurate for identifying inherited grains than manually removing analyses by visual inspection. This is particularly relevant for Paleoproterozoic rocks, where a conservative 1% uncertainty corresponds to approximately ± 20 Ma, leading to inevitable overlap between crystallization ages and slightly older xenocrysts (e.g., see comments on sample GLR-35 below).

Some comments on the data of a few samples:

Sample LA-01:

- Grains LA01_17 and 18 overlap with the main group and, therefore, should not be considered as inherited. They should be included in the weighted mean $^{207}\text{Pb}/^{206}\text{Pb}$ and concordia age calculations. Based on the rejection criteria indicated in the supplementary table (concordance > 0.95), grain LA01_10 could also be included in the weighted mean $^{207}\text{Pb}/^{206}\text{Pb}$ age calculation.
- Grain LA01_19 (2064 ± 19 Ma) should be considered as an inherited grain as it aligns with LA01_3 (2061 ± 23 Ma) and 13 (2061 ± 22 Ma).
- Grains LA01_20, 24 and 25 seem affected by common Pb.

We recalculated the ages after your recommendations, MSWD raised to 2 and the age of the entire population became older. By using the radial plot, we obtained one tail with 1971 ± 13 , which is geologically more meaningful for a crystallization/eruption age. This is now explained in the text and the figures were replaced in the Supplementary Table 3.

Sample PA-34:

The reported MSWD of 3.4 the weighted mean concordia age ($2002.3 \pm 3.2/6.3$ Ma) exceeds the range of acceptable MSWD values for $n = 19$ at 2 sigma level. The weighted mean $^{207}\text{Pb}/^{206}\text{Pb}$ age should be the preferred age (1996 ± 4 Ma, MSWD = 0.87).

Ok, this is now fixed.

Sample TM43:

Using the 15 analyses selected by the authors to calculate the weighted mean $^{207}\text{Pb}/^{206}\text{Pb}$ age, I obtain a lower MSWD value of 1.3, compared to the reported 2.5 in the data table. This suggests that the weighted mean $^{207}\text{Pb}/^{206}\text{Pb}$ age should be preferred over the weighted mean concordia age.

This is probably because the analytical errors in our data table are reported as 2σ (absolute), so all diagrams were generated using the ‘Input errors’ setting as ‘ 2σ (abs)’ in IsoplotR.

Sample MA-02:

Similarly to the sample above, using the 22 analyses selected by the authors to calculate the weighted mean $^{207}\text{Pb}/^{206}\text{Pb}$ age, I obtain a lower MSWD value of 0.56, compared to the reported in the data table. In any case, both of these MSWD values are statistically more reliable than the MSWD of 4 reported for the weighted mean Concordia age, which was chosen as the preferred age by the authors. Given this, the weighted mean $^{207}\text{Pb}/^{206}\text{Pb}$ age should be preferred over the weighted mean Concordia age.

Ok, this is now fixed.

1.6

Sample GLR-35:

- The statistical parameters of the three “age” types provided clearly indicated that the calculated dates are not statistically sound:
 - concordia age (n = 23): MSWD = 6.8 and p = 0 (for concordance and equivalence);
 - weighted mean 207/206 age (n = 23): MSWD = 4.8, p = 0);
 - weighted mean concordia age (n = 20): MSWD = 3, p = 0).

The three analyses, considered inherited by the authors, are represented in grey. However, both the MSWD values and visual inspection of the diagram clearly indicate the presence of more than one population of analyses within the green group.

- Considering that the 23 analyses are all nearly concordant, we can use the radial plot function of Isoplot-R to assess the presence of more than one age component. The algorithm indicates the presence of two distinct age components within the 23 analyses, one at **1795 ± 6 Ma** and an older one at **1830.6 ± 8.9 Ma**. From the plot, it is clear that analyses older than 1820 Ma belong to the older component.
- Considering analyses older than 1820 Ma as inherited (GLR35_8, GLR35_16, GLR35_19, GLR35_22, GLR35_24, in addition to the 3 selected by the authors), we can calculate a weighted mean 207/206 age of 1795.2 ± 5.6 Ma (MSWD = 0.82, p = 0.66) for the younger group.
- Using the MLA algorithm, we obtain an identical age of 1794 ± 7 Ma.

Ok, we used the younger tail in the radial plot as the preferred age.

Radial plot (finite mixtures = auto)

Radial plot (finite mixtures = minimum)

Sample PGR27:

- The authors consider only the existence of one inherited grain (PGR27_25). However, inspection of the Concordia diagram clearly indicates the presence of more inherited grains. Excluding analyses PGR27_8 and PGR27_9, which have very large uncertainties, it is clear that there are at least four analyses that are older than the main group.

- This is clearly depicted in the radial plot, which shows a younger population with a weighted mean $^{207}/^{206}$ age of 1791 ± 5 Ma and an older one at 1840.4 ± 9.9 Ma.

Ok, we used the younger tail in the radial plot as the preferred age.

central age = 1799.9 ± 8.2 Ma (n=26)
MSWD = 3.7, $p(\chi^2) = 3.8e-09$
dispersion = $0.95 \pm 0.39\%$

Sample SS-39:

Rejecting analyses SS39A_23 and SS39A_25 results in a statistically more robust concordia age (considering MSWD and p for concordance and equivalence).

Ok, we used this concordia instead of the earlier one.

Sample PB07A:

- Analyses PB07A_18 and PB07A_24 are not represented in the diagrams and were not used in the calculations; however, this is not indicated in the data table.
- From the inspection of the Concordia diagram, it is clear that, apart from grain PB07A_8 (2852 ± 17 Ma), there are other xenocrysts. This is also evident from the statistical parameters reported by the authors for the Concordia age, where MSWD = 5.1 and $p = 0$ (for concordance and equivalence) clearly point to excess scatter. Indeed, the radial plot indicates the presence of an older group comprising 4 analyses with a weighted mean 7/6 age of 1814 ± 12 Ma, and a younger group, the main one, at 1762.2 ± 5.4 Ma. Calculating a Concordia age for the latter group yields an excellent age of 1761.3 ± 3.9 Ma (MSWD = 1.1, $p = 0.28$, for concordance and equivalence).

This does not differ very much from our weighted mean concordia age of 1760.8 ± 3.3 (MSWD = 1.4, N = 17, $p = 0.37$).

Sample PB09A:

- This is a sample where the differences between the plots constructed for the same analyses with the data reported in the supplementary table (left) and the plot presented by the authors (right) differ significantly (see figures below). From the inspection of the diagram and the 7/6 ages (1810 ± 21 Ma and 1796 ± 22 Ma), it is clear that the two analyses in red on the left diagram are older than the main group (ages around 1760 Ma). This difference is not depicted in the diagram presented by the authors (right), which raises questions regarding the validity of the plots.
- As I mentioned before, while this does not have major implications for the calculation of the crystallization ages of the studied samples, since the final age differences overlap within uncertainty and therefore do not affect the conclusions of the study, it does indicate that something is wrong. While I can't be 100% sure that this is the cause of the issue, my guess is that it has to do with the fact that users plotted the diagrams using the 207/235 (not reported in the data table) and 206/238 ratios, meaning that the results calculated by Lolite are producing slightly inconsistent isotopic ratios (this is a known issue of Lolite).

- Considering the data presented in the data table, which is the one I have access to, this sample as an inherited component with a WM 7/6 age of 1803 ± 16 Ma and a crystallization age of 1758.6 ± 5.4 Ma (WM 7/6 age of the youngest group); the latter overlap with the Concordia age of 1758 ± 4 Ma (MSWD = 1.4, $p = 0.07$; excluding the most discordant analysis PB09_21).

Ok, we recalculated the ages and presented it in a revised table with the diagrams (Supplementary Table 3).

concordia_age::: 1758 ± 4 Ma (n:17)

MSWD: 0.0041 μ 1.4 | 1.4 | σ 0.95 | 10.058 10.07 2

Response to Referees
**Large-scale crustal growth determined by LIP magmatism during the
Paleoproterozoic**

Matheus S. Simões, Carlos A. Sommer, Marcelo L. Vasquez, Lucas M.M. Rossetti, John M. Cottle, Túlio A. Mendes, Andrew R.C. Kylander-Clark

Response to reviewer 2

Reviewer #2 (Remarks to the Author):

The main constraint of the manuscript is the lack of tectonic evidence, among others, compatible with SLIP setting for the study regions. Sometimes, only geochemistry data

lead to dubious conclusions. According to the chemical composition of the rocks (mainly calc-alkaline to shoshonitic, high-K, I-type to A-type), it is possible to accept these rocks as occurring in pre- to post-collisional, as well as in an intraplate (fissural) environment. In fact, several authors argue magmatic arc setting for the rocks included in the three cycles proposed in the manuscript. It is the case of Fraga et al. (2020, 2024) for the Orocaima Igneous Belt (cycle 1); Almeida et al. (2001), Vasquez et al. (2002), Santos et al. (2004), Juliani et al. (2013), Fernandes and Juliani (2019) for the Tapajós and Irixi-Xingu domains (cycles 1 and 2), and Cordani and Teixeira (2007) for Juruena region (cycle 3).

In the manuscript, the SLIP-related setting is accepted for all regions without discussion. Arc magmatic and intraplate events may simultaneously or sequentially happen, since well-arranged in the space and time. For instance, in the Tapajós domain, where rocks from cycles 1 and 2 occur, and also magmatic arcs are argued, it is necessary reconstruct the tectonic history, fit both cycles and prove if they are true SLIPs. In summary, I am not against the SLIP hypothesis, but it is necessary show evidence in a comprehensive geological setting. They could be shown in tables or as Supplementary Material.

To provide further context, we have included a new supplementary table in the revised manuscript that summarizes arguments supporting or challenging both LIP- and subduction-related models.

We fully agree with the importance of seeking balanced interpretations regarding the Amazon Craton (AC). Naturally, accretionary and collisional events played a role in the AC's tectonic evolution, and we do not dismiss their occurrence. As highlighted in the new supplementary table, several regions of the AC with Rhyacian to Statherian ages now present stronger evidence for intraplate magmatism, even in areas that were initially described as magmatic arcs. The historical view of geochronological provinces is a hallmark for understanding crustal evolution of the AC. Orogenic model-based solutions were a solid response to primarily geosyncline theory-based models. Nevertheless, there were some flaws adopted for reaffirming these models. However, some key assumptions used to reinforce these models warrant reconsideration.

One major limitation lies in the interpretation of whole-rock geochemistry:

1. Most silicic rocks are naturally enriched in K_2O and plot within the high-K to shoshonitic fields on petrological diagrams. So, not an argument for magmatic arc setting.
2. These diagrams have often been used to support Andean-type magmatism without incorporating A-, I-, or S-type granitoid classification diagrams and without considering the geochemistry and evolution of coeval mafic magmatism.
3. A significant volume of these high-K silicic rocks is demonstrably A-type in character, as already established in previous studies, and therefore does not require detailed demonstration in this manuscript.
4. The common association of I-type calc-alkaline or tonalite-granodiorite-diorite suites (contemporaneous to A-type magmatism) with arc settings has often been misapplied. The main misconception regards the lack of detailed geological mapping. For example, single plutons in intraplate settings may record multiple intrusive pulses, alternating between syeno- to monzogranites with A-type affinity and granodiorites/tonalites with I-type affinity—as observed,

for example, in the Lavras do Sul (Gastal et al. 2015 - 10.1590/23174889201500020004) and São Sepé (Gastal and Ferreira, 2013 - 10.22456/1807-9806.43440) granites of the Neoproterozoic Sul-Riograndense Shield.

5. Furthermore, many igneous suites—coeval with known A-type magmatism—consist of high-K, calc-alkaline I-type granitoids, which are not exclusive to arc settings, and can also be generated by high-grade metamorphism of deeply buried continental crust and subsequent attendant water-fluxed melting of crustal lithologies (Janousek et al., 2023 - 10.1016/j.jog.2022.101960).

A second flawed assumption involves the interpretation of age zonation in zircon U–Pb data and Nd isotopic signatures as indicative of accretion. The first interpretation of Cordani et al., Tassinari and Macambira, Santos et al., Cordani and Teixeira, assume isotopic zonation across the (generally) NW-trending geochronological provinces. Nevertheless, no clear age/isotope ratio zonation is observed for Cycle 1, which is dominantly E-W trending in the Guyana Shield. For cycle 2, there is no age zonation, and the Nd isotopic signatures are evidently influenced by underlying Archean or Rhyacian–Siderian crust (Semblano et al., 2016 - 10.11606/issn.2316-9095.v16i3p19-38; Teixeira et al., 2017 - 10.1016/j.jsames.2017.09.017, and many references from works at eastern AC). Cycle 3 is also E-W trending and exhibits strong evidence for extensional, bimodal magmatic activity and coeval ultra-high temperature (UHT) metamorphism (Rizzotto et al., 2019 10.1016/j.jsames.2019.01.003, Benetti et al., 2024 - link). Nd isotopes and inherited zircons points to a reworked Orosirian crust with local Archean substrate. Later deformation and metamorphic overprints correspond to younger tectonic events unrelated to the timing of LIP emplacement (Almeida et al., 2022 - 10.29396/jgsb), as illustrated in Figure 6.

Another discussion needing to be enlarged concerns the extensions and the representations of the three cycles in Figure 1. In the Tapajos and Iriri-Xingu domains (southeastern part of the craton), rocks from both cycles 1 and 2 occur. In my point of view, it is not possible separate two different domains, as it is shown in Figure 1, because in some areas there are both cycles, side-to-side, overlaying or cross-cutting one over the other. The same is true to Peixoto de Azevedo domain, surrounded by rocks from cycle 3, where rocks from cycle 2 is also present. Additionally, a 1.88 Ga granite magmatism is spread in the Carajas domain, extreme southeastern part or the craton. This magmatism has been correlated to Uatuma SLIP, present in the adjacent Iriri-Xingu and Tapajos domains. In line 331, a rock from cycle 2 occurring in Tucuma area (Carajas Province) is exemplified, but it is not shown in figure 1. I suggest including as Supplementary Material a map presenting all sample locations, which data were used in the manuscript.

We acknowledge that several aspects of geological mapping may be limited by the scale of the maps presented. Nevertheless, we carefully reviewed all the points raised here and confirmed that they are indeed represented on the geological map. To improve clarity, we have included a new Figure 1 in the revised manuscript, where the geological outlines are more precisely constrained. Additionally, detailed geological maps showing sample locations and descriptions are now provided in the Supplementary Data 1 file.

Reviewer #2 (Remarks to the Author):

The reviewed manuscript “Large-scale crustal growth determined by LIP magmatism during the Paleoproterozoic”, was evaluated. The manuscript has been substantially improved, especially with the inclusion of new maps and supplementary material. Now, it is clearer, precise and denser. However, I have some additional comments, hoping the manuscript to be better.

- I suggest including in the title “Amazonian Craton”, since the object of the manuscript is exclusive over this region.

- Indeed, in my first review, in according to Stratigraphic code, I suggested replacing “Amazon Craton” by “Amazonian Craton”, term originally proposed by Almeida et al. (1981) to define the same region and setting. This point was not commented by the authors.

We apologize for not addressing this point earlier.

The term **Amazonian Craton** was first proposed by F.F.M. Almeida and colleagues in their landmark 1981 paper. In my view (first author), Almeida remains one of Brazil’s most influential geologists. His earlier work in the southeast Amazon (Almeida & Nogueira Filho, 1968) was fundamental for clarifying the region’s complex geology along the Aripuanã river, which was often misinterpreted by later studies that overlooked his contributions. His research on the Trindade and Fernando de Noronha islands continues to inform interpretations of volcanic architecture, and several of his other works remain foundational for understanding Brazil’s geology. I highlight this background to express my deep respect for his legacy.

A later review by Santos (2003) argued—convincingly, in my view—that we should avoid adjectives when naming tectonic units, otherwise we would end up with terms such as “Sanfranciscan Craton,” “Kaapvaalian Craton,” “Superiorian Craton,” or “Yilgarnian Craton.” The **International Stratigraphic Code** (Murphy & Salvador, 2008; <https://stratigraphy.org/guide/defs>) does not provide guidance on this issue either, since it applies to litho-, bio-, chrono-, and magnetostratigraphy, not to tectonic nomenclature.

In practice, the literature shows no consensus or problem: some international papers use *Amazonian*, others *Amazon*. For these reasons, I chose to retain the usage in our manuscript. I acknowledge the reviewer’s concern, but I believe it is a matter of convention rather than correctness, and I trust the editors will consider it within the broader scientific context as referees.

- The new “Arc vs LIP” table summarizes a comprehensive characteristic set helping the discussion about the tectonic setting. However, it is possible to see that they are not conclusive, in special, concerning poorly known regions, as we have in the

Amazonia Craton. For instance, considering this constraint, the debate Arc vs LIP is not a crucial obstacle to advance the discussion. The main aspects to support the manuscript approach are the huge magma volumes with age, isotope and chemistry compositions nearly well determined.

We understand that the table, prepared following a suggestion, is not definitive—particularly given uncertainties in poorly known regions. However, it is informative and helps readers draw their own conclusions. Nevertheless, we are open to suggestions and, if requested, we can delete this from the Supplementary Material.

- Concerning this aspect, I believe that the statement "...since no plate tectonic indicators have been identified" (line 71) is too strong to the actual level of geological knowledge of the Amazonian Craton.

Although geological knowledge of remote Amazonian regions is limited, this has not prevented the proposal of diverse tectonic models. Our interpretation of PTIs relies on major advances over the last two decades—airborne geophysics, geochronology, geochemistry, isotope studies, and fieldwork—that have revealed features such as ophiolite sequences (Rizzotto & Harmann, 2012). Naturally, future discoveries of plate tectonic indicators for the 2.0–1.74 Ga interval may lead to revised interpretations.

- In Fig. 1, why were the Paraguá and Rio Apa domains excluded from the Amazonian Craton? Most authors accept Paraguá domain as allochthonous block, but it is not a reason to exclude it. For example, the Archean Imataca and Amapá blocks probably are also allochthonous, but were considered as part of the craton.

We have no problem with Paraguá and Rio Apa, including these units in the geological map would not best illustrate the LIP–SLIP units and dyke swarms concentrated in the main area of the Amazon Craton.

So, understanding that the way it is shown in the Figure Caption caused a confusion, we rephrased that: "*(B) Chronostratigraphic map of the Amazon Craton (the southern Paraguá and Rio Apa domains are not shown in this figure, as their inclusion would alter the map scale and reduce the detail of the LIP–SLIP units and sample locations)*"

- In relation to the geochronological provinces, it is important to highlight that not all of them are considered as orogenic belts (line 132). It is the case of the Carajás domain, as well as the Irixi-Xingu and Erepecuru-Trombetas domains, all of them included in the Central Amazonia Province. Moreover, the NW-trend Irixi-Xingu and Erepecuru-Trombetas domains are almost fully composed by rocks from cycles 1 and 2.

We agree with the reviewer. The original sentence was not adding anything to the paper, so we deleted it.

- Still concerning the geochronological provinces, not all of them are NW-trending orientation (see Response for Reviewer 2). It is true for provinces younger than 2,0 Ga, but it is not the case of the Maroni-Itacaiunas Province (Guiana Shield).

We understand this may arise from the former phrase we have deleted. Nevertheless, Maroni-Itacaiúnas, Ventuari-Tapajós, Rio Negro-Juruena, Rondonian-San Ignácio and Sunsás are NW-trending (we can call it SE-trending also, but I think it is the same), as seen below (from Cordani et al., 2009):

- I disagree that the "...the Orosirian-Statherian orogenic belts do not match the orientation of N-S and E-W Orosirian-Statherian igneous belt" (lines 132 and 133). Cycles 1 occurs in the E-W Maroni-Itacaiunas Province, as well as in the NW Central Amazonia Province, this last one completely composed of rocks from the cycles 1 and 2. Cycle 3 crops out in E-W southern part of the Rio Negro-Juruena Province.

However, it doesn't matter if the igneous and orogenic belts match or not. The SLIP magmas could utilize previous regional structures to penetrate the upper crust.

We agree with the reviewer. The original sentence was not adding anything to the paper, so we deleted it.

- In regard to lines 533-535, there are several authors reporting evidence of plate tectonics in the Rio Negro Province (e.g. Almeida et al. 2022 (<https://doi.org/10.1080/00206814.2021.2025158>), and reference therein.

This is already cited in the paper: '*A potential exception is the proposed Statherian arc magmatism in the Rio Negro Province¹⁶; which lack definitive plate tectonic indicators but also lack positive evidence for the absence of plate tectonics.*', but we added the reference.

- Reference 340 (line 252) and fig. 2d (line 269) do not appear in the manuscript.

This was a misprint and should be fig. 3d. Now fixed, thanks.

General comments

The manuscript convincingly argues that LIPs were major contributors to Paleoproterozoic crustal growth. However, has this not been shown previously by for example the work of reference 12–14?

We agree that refs. 12–14 are landmarks for recognizing LIP activity in the Amazon Craton, but none develops a craton-scale synthesis arguing that LIPs were the primary drivers of Paleoproterozoic crustal growth. Specifically: (12) Klein et al. (2012) first recognized a silicic LIP at 1.88–1.87 Ga but treated a subset of the record and did not evaluate crustal growth at craton scale; (13) Rizzotto et al. (2019) reinterpreted the southwestern Amazon Craton and shifted the discussion away from purely accretionary models, yet it was not intended as a craton-wide treatment of Statherian magmatism; (14) Ibañez-Mejía et al. (2025) documents a ca. 1.98 Ga LIP, focusing largely on mafic magmatism rather than silicic components and without a continent-scale crustal growth framework.

In summary, these three works are hallmarks, but they do not integrate craton-scale models and do not argue in favor of LIPs as crust builders during the Paleoproterozoic.

The idea of mantle cyclicity at ~95 Myr intervals aligns well with emerging models of deep Earth dynamics. (Just be honest with the ages here)

It places the Amazon Craton in a global context through correlations with West Africa, Baltica, and others—strengthening the case for a broader LIP-linked geodynamic system.

I am a bit puzzled why the authors only use, and base their discussion on, Nd model ages, when their own data is zircon Hf data and model ages can be calculated from these data. Is there a reason for this?

We prefer not to use Hf TDM ages because it has been demonstrated in several papers it can produce artificial age ranges depending on the chosen source.

Bea et al. (2018) - <https://doi.org/10.1016/j.chemgeo.2017.11.034>

Veervort and Camp (2016) - <http://dx.doi.org/10.1016/j.chemgeo.2016.01.023>

The authors claim to have calculated two stage model ages (although undefined) but do not present them or seem to use them.

Two stage model ages were calculated for literature Nd isotope data and the calculations were made by the referred authors. No model ages were used for Hf in this paper.

The abstract could benefit from more clarity by directly stating the key finding: that LIPs contributed significantly to crustal growth via deep-crustal partial melting.

We agree, now revised.

The discussion could be better segmented to explicitly highlight contrasts between arc and LIP-related crust formation models.

Please, take a look at the first two paragraphs of the last section. We improved the LIP crust formation models and avoided flooding the paragraphs with detailed explanations which are already (and very well) discussed in the literature.

While the authors argue for intraplate LIP-driven crust formation, the paper could further address counterarguments from proponents of arc-dominated growth models.

This represents a case of “*difficult ground*” (Sun Tzu), as one of the reviewers is presumably from Brazil. To address this, we added a paragraph drawing on Supplementary Table 2 (Arc versus LIP), but we avoided an extensive discussion so as not to risk overemphasizing potentially contentious interpretations.

‘Earlier interpretations of subduction episodes following the Transamazonian–Birimian Orogeny were based on arguments such as juvenile Nd signatures in granitoids and gneiss²⁶, calc-alkaline geochemical signatures in tonalites and monzogranites (Santos et al., 2004), “porphyry-like” gold deposits (Juliani et al., 2021), and regional zonation of crystallization/metamorphism ages coupled with Nd model ages¹⁶. Nevertheless, many of these features are also found in Phanerozoic silicic LIP provinces (Ewart et al., 1992; Bryan, 2007; Camprubí et al., 2017). In contrast, diagnostic features such as large-scale radial dyke swarms, bimodal igneous compositions, and the short-lived nature of voluminous magmatism across vast regions are not typical of arc-related settings (Ducea et al., 2025).’

The interpretation of zircon trace elements as direct indicators of source pressure and temperature, though accepted, would benefit from more explicit discussion of uncertainties and assumptions.

We also felt this section needed some improvement. Now this is slightly less interpretative when describing the data and we added two references that corroborate with field evidence of the sources (HT-UHT garnet-bearing granulite/migmatite).

Compared to zircons from high-temperature, high-pressure metamorphic rocks equilibrated with garnet, the igneous zircons analyzed here display $Yb_{(n)}$, $Yb/Ce_{(n)}$ and $Yb/Tb_{(n)}$ similar to those of lower crustal material³⁷(Fig. 3a, b). This source interpretation is consistent with field evidence indicating that melts for cycles 1 and 3 were generated from HT-UHT garnet-bearing migmatites and granulites at lower continental crustal levels^{38,39}.

Assuming a pressure gradient of 3.5 kbar/km, these pressures corresponded to crystallization depths of ~29 km for Cycles 1 and 2 and ~27 km for Cycle 3, consistent with field evidence and zircon trace-element data indicating sources within the lower continental crust.

Comments on data

Firstly, we would like to thank you for the detailed revision of the original data. Had this not been done, and the paper eventually accepted, it would have lost credibility. Your

review gave us the opportunity to demonstrate the robustness of our data in an organized and transparent way.

Secondly, we apologize for the problems with the dataset. The main issue stemmed from an incorrect reorganization of the original spreadsheet, which resulted in several row offsets and confusion between reference material data. Also, there was a problem with the methods section which is now fixed.

To address this, we carefully traced the dataset back to the original spreadsheet and restarted the review process from the beginning. We believe this has resolved the inconsistencies, and we now proceed to respond to the reviewer's comments in detail below.

I cannot recreate the ϵ_{Hf} values for the data.

The authors do not define what values for, for example CHUR was used. (Bouvier et al., 2008)?

This is now indicated in the methods section. We used Bouvier et al. (2008) values.

$$\epsilon_{\text{Hf}} = \left(\frac{{}^{176}\text{Hf}/{}^{177}\text{Hf}_i}{(0.282785 - 0.0336 * (\text{EXP}(0.01867 * 0.001 * \text{AGE}) - 1))} - 1 \right) * 10000$$
$${}^{176}\text{Hf}/{}^{177}\text{Hf}_i = ({}^{176}\text{Hf}/{}^{177}\text{Hf} - ({}^{176}\text{Lu}/{}^{177}\text{Hf} * (\text{EXP}(0.01867 * 0.001 * \text{AGE}) - 1)))$$

Using the same decay constant and bouvier CHUR values I get quite substantially different ϵ_{Hf} values than the authors. This is VERY concerning! Even using other (common) CHUR values I doubt I would get the same values as the authors.

After considering this comment, we recalculated all ϵ_{Hf} values using the parameters of Bouvier et al. (2008). The results differed by only 0.1 epsilon units from our previous calculations. Since this matches what the reviewer should obtain, we are confident that this issue is now resolved.

In several places (for example row 28 (VR05-1)) there are Hf/Hfi data and a calculated ϵ_{Hf} value. But for that analyses (and several more) there are no raw data in columns CF–CO. The table is generally very confusing and difficult to decipher. ϵ_{Hf} values are calculated for zircon with no data and vice versa.

We apologize for the errors. When preparing the table for the manuscript, some data became offset from the original spreadsheet. In addition, the original dataset itself contained a few mismatches (notably one involving OGC and Tem2). We have carefully reviewed and corrected these issues, and we believe the dataset is now accurate.

Even though not stated the ϵ_{Hf} values seem to be calculated at the individual zircon dates (based on spot locations on figure 4). This has been shown multiple times that this is in accurate. See for example Whitehouse and Kemp (2010, Geol. Soc. Spec. Pub. London).

ϵ_{Hf} values should always be calculated at the interpreted age of igneous crystallisation.

We agree with this comment and have therefore recalculated all individual ϵ_{Hf} values using the interpreted igneous crystallization ages. For grains interpreted as inherited, however, we retained their individual zircon $^{207}\text{Pb}/^{206}\text{Pb}$ ages. This is now explained in the methods section.

The “preferred age” for several samples according to the summary table on tab Sample identification state that they prefer and use the “concordia weighted mean age”. What is that? Either it’s a concordia age or a weighted mean age. And if a weighted mean age, then it needs to be defined as for example $^{207}\text{Pb}/^{206}\text{Pb}$ or similar. If this is a weighted mean of individual Concordia dates (which is unconventional, but would be ok) this needs to be explained, and the individual concordia dates should be added to the data column in the supplementary table.

The only form of explanation is found on lines 201–202: “Ages are reported here as the maximum likelihood estimate with its analytical uncertainty/analytical uncertainty with dispersion.

This problem originated from our initial interpretations and was further complicated by a reviewer’s suggestion to use radial plots. We have now adopted the weighted mean of $^{207}\text{Pb}/^{206}\text{Pb}$ dates as the most reliable age estimate. This choice is supported by the high accuracy of the $^{207}\text{Pb}/^{206}\text{Pb}$ age of OG reference material. The Supplementary Material has been updated accordingly.

Furthermore: 6/7 weighted mean in Supp. table 2. Tab “Sample identification” should be 7/6 I assume.

Ok, corrected.

OGC $\epsilon_{\text{Hf}} = +0.6 \pm 0.2$ 2s (Kemp et al., 2017), in this study the results come out to: ~+2.7- 2.5 units too high. Is this data correct?

Some reference zircon have data for Hf in column T–W, but some only in row CF and forward. What is the difference here? OGC have no calculated ϵ_{Hf} values, but there is raw data. It’s all very confusing, and the fact that these reference zircon are giving the incorrect values combined with the unreadable table makes me worried.

(However, the data in row CF in the “data” tab does not seem to be the measured values, as this doesn’t make sense for any of the reference zircon) This needs to be marked more clearly so that one can check the data quality of the reference material.

The issues with OG and the overall confusion involving row offsets and mismatched or missing reference material data stemmed from the incorrect reorganization of the original spreadsheet and from a lack of clarity (and some misprints) in the methods section.

When we performed the Hf analyses, we used different reference materials than for the U-Pb analyses. In some cases, the same material was analyzed for both systems, but others were exclusive to one technique (e.g., OG was used only for U-Pb, while Mud Tank was used only for Hf). This distinction is now clearly explained in the methods section and the Supplementary Material spreadsheet has been reorganized to properly reflect these datasets.

“To assess accuracy and precision, natural (91500, ‘Mud Tank’, and GJ-1) and synthetic (MunZirc-1, -3, and -4) reference zircons were run concurrently and treated as unknowns.”

This data is nowhere to be found! At least not for Mud Tank and MUN zircon.

Also caused by imprecise information in the original methods section. To be crystal-clear:

U-Pb was performed on GJ-1; OG1; Plešovice and Temora-2

Lu-Hf was performed on 91500, GJ-1, Plešovice, Mud Tank and Temora-2.

This is now explained in the methods section (with obtained ages and isotope ratios) and organized in the Supplementary Material spreadsheet.

What lab was used for these isotope analyses? The machines are listed, but not what lab.

How was the primary standard values for zircon Hf data checked? Against JMC-475? Not reported. Also, the $^{178}\text{Hf}/^{177}\text{Hf}$ is higher than usual in these analyses. A mean value of all analyses is 1.4677 when the reported value is 1.4772 for, for example JMC-475. All individual analyses have a $^{178}\text{Hf}/^{177}\text{Hf}$ > than 1.4772. Please comment on this.

We inserted the LASS-UCSB in the methods section.

Yes, the mean value is for $^{178}\text{Hf}/^{177}\text{Hf}$ in our dataset is 1.46773 and the median is 1.46760. However, we disagree that the accepted value for JMC-475 is 1.4772, as it has been reported as 1.467254 (e.g., Choi et al. 2013 - <https://doi.org/10.1186/2093-3371-4-1>). Our average differs by ~0.0004 from this reported value, most likely due to minor gain calibration effects.

For the other Hf isotopic ratios, we normalize the Plesovice zircon. The $^{178}\text{Hf}/^{177}\text{Hf}$ ratio, however, is internally standardized and therefore not normalized to Plesovice. Its slight deviation from the accepted value is attributed to the gain calibration offset. Importantly, the accuracy of our $^{176}\text{Hf}/^{177}\text{Hf}$ ratios is confirmed by the secondary reference materials (91500, Mud Tank, Temora-2, and GJ-1), all of which fall within uncertainty of their accepted values.

What was used in the Yb correction? Among the data presented in the table there is no zircon with high Yb which would be usable for Yb corrections. OGC would be good. Mud Tank would be good. MUN would be ok. But none of this data is presented.

We use Plesovice zircon to correct for Yb interference (and any associated offset in Hf). As demonstrated by the reference materials with strongly contrasting Yb contents (e.g., Temora-2 with >150,000 ppm Yb, and GJ-1 with ~34,000 ppm Yb), the corrected $^{176}\text{Hf}/^{177}\text{Hf}$ values fall within uncertainty of their accepted values. Reference materials for low $^{176}\text{Yb}/^{177}\text{Hf}$ were Mudtank, GJ-1 and Plesovice. For high $^{176}\text{Yb}/^{177}\text{Hf}$ we used 91500 and Temora-2.

Individual line comments

Line 14: The rise of plate tectonics in the Paleoproterozoic. A LOT of studies would

argue against this, but here its stated as a truth. This needs to be addressed.
We agree. Replaced 'rise' by 'consolidation'.

Line 65: There are basically no 2.2 Ga rock in Fennoscandia. To find 2.2 Ga old bedrock in the region, you only find it in the eastern and northern parts, specifically within the Karelian Craton, in areas of Finland and Russian Karelia (In this paper the Kola-Karelian) where Paleoproterozoic supracrustal sequences, often associated with early rifting events, are preserved.

Ok, Fennoscandia is now excluded from this phrase.

Lines 74–75: Why 95 Myrs.? The timespan between these magmatic episodes are 100, 100 and 110 Ma.

Ok, $1980 - 1880 = 100$ Myr and $1880 - 1790 = 90$ Myr.

Replaced by 'Peaks in magmatic activity within these provinces occurred at approximately 90 to 100 Myr intervals'

Line 158: Is negative Nb-Ta anomalies really likely to be present in LIPs? You need either significant crustal contamination during magma ascent and emplacement, or fractionation of Ti-bearing minerals within the magmatic system. I would rephrase this sentence.

Apparent arc signatures, such as negative Nb-Ta anomalies are very common, mainly in low-Ti tholeiites from LIPs. There is a sub-chapter in Ernst (2014) – Geochemistry of LIPs (<https://doi.org/10.1017/CBO9781139025300.010>), resuming additional causes for it:

(1) partial melting of heterogeneous subcontinental lithospheric mantle (SCLM; e.g. Gallagher and Hawkesworth, 1992; Jourdan *et al.*, 2007a, 2009);

(2) partial melting of variably depleted and hydrated mantle, inferred to be within the lithosphere (Turner and Hawkesworth, 1995);

(3) partial melting of lithospheric mantle that has previously been modified by subduction-related fluids (e.g. Sandeman and Ryan, 2008);

(4) partial melting of the lithospheric mantle that had been modified by subduction processes during major crust-formation events in the region (Zhao and McCulloch, 1993);

(5) contamination during passage through the SCLM of magmas derived from sublithospheric plume or MORB sources, or mixing between sublithospheric and lithospheric mantle melts (e.g. Marsh and Eales, 1984; Arndt and Christensen, 1992; Riley *et al.*, 2005; Jourdan *et al.*, 2007a; Heinonen and Luttinen, 2010);

(6) assimilation of sediments (Elburg and Goldberg, 2000);

(7) polybaric fractional crystallization (Marsh and Eales, 1984);

(8) contamination by sedimentary country rocks combined with fractional crystallization in addition to processes at depth (e.g. Riley *et al.*, 2005, 2006; Jourdan *et al.*, 2007a);

(9) contamination of mantle melts by crustal material in deep–mid-level crustal “staging chambers” (e.g. Nelson *et al.*, 1990; Ihlenfeld and Keays, 2011; Ciborowski *et al.*, 2013; Jowitt and Ernst, 2013);

(10) crustal contribution derived from within an ancient region of the mantle lithosphere, i.e. from recycled sediment rather than from the overlying continental crust (Lightfoot *et al.*, 1993b); or

(11) sudden reactivation of dormant arc or backarc sources trapped under continental-plate sutures (Puffer, 2001).

Line 209–: Results are written in past tense (yielded) but these analyses still yield. Hence, this should be in present tense (yield).

Ok, changed all yielded to yield in the results section.

Line 231: Add these ratios as columns in supplementary table to easily check. A mean Th/U of 1 is very high.

After a careful examination of Th/U ratios in zircon, the compilation of Kirkland *et al.* (2015; <http://dx.doi.org/10.1016/j.lithos.2014.11.021>) indicates that igneous zircon has a mean Th/U of 0.77 (N = 10,693). For specific rock types, mean values are 0.85 in alkali granite (N = 280), 1.03 in diorite (N = 249), 0.84 in granite (N = 1963), and 0.98 in quartz monzonite (N = 102). Thus, a mean Th/U close to 1 is not anomalously high; it is slightly above the average for granitic compositions but well within the expected range for diorite and quartz monzonite. Nevertheless, as requested, we have included the ratios as a column in the supplementary table.

Fig. 1. Histogram of Th/U ratio of magmatic zircon analysed by SIMS.

Also, the word “ratios” is redundant as the Th/U and Yb/Sm are ratios. Hence, this reads “ratio ratio”.

Ok, rewritten.

Lines 291–295: Statement requires references.

Ok, now properly cited.

Line 300: What DM was used? What CHUR values were used?

We added a sentence in the beginning of this section citing “Reference values for the chondritic uniform reservoir (CHUR) are from Bouvier et al. (2008) whereas those for the depleted mantle (DM) are from Andersen et al. (2009).”

Lines 303-304: “Primary sub-populations for Cycle 1, identified in KDE plots, occur at -5.4 and +2.7.” 1. What does this mean? 2. Occur?

Replaced by “Primary sub-populations for Cycle 1, identified in KDE plots, display peaks at -5.4 and +2.7”

Line 305: clustering at -0.89 is very specific, please rephrase.

Replaced -0.89 by -0.9.

Line 306: haspredominantly (missing space)

Ok, fixed.

Line 311: Figure 3? Should be figure 4, right?

Yes, fixed.

Line 315: What is clustering here at the specific value of +0.68? All data? Some of the data? It doesn't look like a tight cluster. If this is a mean value I assume the MSWD is rather high if all data is included. If all data is not included, then this needs to be stated. Oh, now I saw the curve in the figure. So this is a peak, not a cluster... Please explain this more carefully.

Replaced 'clustering' by 'with a peak at'.

Fig. 4: What DM was used in the model age calculations for Nd data?

This is explained in the figure caption: 'Unfilled squares are data from rocks that do not belong to the igneous cycles. DM curve according to [52].'

[52] DePaolo, D.J., 1981. Neodymium isotopes in the Colorado Front Range and crust–mantle evolution in the Proterozoic. *Nature* 291, 193–196.
<https://doi.org/10.1038/291193a0>

Line 444: These are not only zircon U-Pb dates. Baddeleyite have been used to date at least a few of these, which is apparent even from the titles of the references.

Yes, corrected: 'Cycles 1, 2, and 3 have U-Pb zircon and baddeleyite ages of'

Line 446: Well. The time difference between the direct ages of the dykes are 100–110 Myrs. The difference between the zircon peaks, which I assume have an uncertainty that overlaps the baddeleyite/zircon ages of the dykes might be ca. 95 Myrs. But these are peaks, not direct ages. I think the authors needs to be more up front about this. Ca 100 Ma is still a good match to your harmonic mantle cycles, but it would be a stronger argument if based on the direct dates presenting the "actual" age gaps. However, the age interval presented and forwarded in the cited reference [58] is NOT 94 Myrs but 90 Myrs. So, now the difference is going from 95–94 to from 100 (or 110) Myrs to 90, which is not as strong of a match.

The suggestion to be clearer regarding these ages is fundamental. Direct ages of basaltic dyke swarms yield intervals of ~100 Myr and ~90 Myr, whereas KDE peaks of zircon crystallization ages from associated silicic rocks suggest intervals of ~93 Myr (1886–1979 Ma) and ~98 Myr (1788–1886 Ma). This paragraph has now been rewritten as follows:

'The intraplate magmatism in the Amazon Craton occurred at consistent, regular intervals. Basaltic dyke swarms associated with Cycles 1, 2, and 3 have U-Pb zircon and baddeleyite ages of 1980 Ma¹⁵, 1880 Ma³¹, and 1790 Ma²⁰, respectively, pointing to 100 Myr and 90 Myr intervals, that potentially represent episodes of mantle partial melting. Contemporaneous silicic rocks yield zircon-age peaks at 1979 Ma, 1886 Ma, and 1788 Ma, as shown in KDE plots (Fig. 7a), corresponding to 93 Myr and 98 Myr intervals, representing peaks of zircon production by partial crustal melting.'

The offset between basaltic and silicic ages could be discussed throughout the paper but may not add much for the reader. In general, ages of basaltic dykes are more precise (typically dated by TIMS, SHRIMP, or ID-TIMS) and may better record mantle melting episodes. Silicic units, in contrast, are more numerous but commonly dated by methods with larger uncertainties (e.g., LA-ICPMS), producing broader peaks that reflect widespread crustal melting during LIP activity.

Regarding mantle cycle periodicity, Condie & Puetz (2019) used time-series analysis and reported that ~50% of age peaks coincide with a predicted ~94 Myr cycle (a simplification of the original 93.5 Myr value). Later, Condie et al. (2021) incorporated ~400 additional samples and concluded that, from 2300 Ma onwards, spectral analysis of LIPs and zircon ages reveals ~24 repetitions of a 90–93 Myr cycle, which they summarized as a “90 Myr cycle” driven by bottom-up mantle processes.

Thus, two sources of uncertainty exist: (i) which reference value to use (94, 93.5, 93, or 90 Myr?), and (ii) which data are most appropriate for comparison (direct basaltic dyke ages defining ~100–90 Myr intervals, or KDE-derived silicic peaks yielding ~98–95 Myr intervals?). Although the correspondence is not exact (but is also not a large discrepancy, like 80 Myr or 120 Myr intervals), the observed ~100–90 Myr recurrence aligns reasonably with proposed mantle-related cycles.

Line 452: This is not ref [58] but rather the work of: S.J. Puetz, K.C. Condie
Time series analysis of mantle cycles Part I: The geologic record in zircons, large igneous provinces and mantle lithosphere *Geosci. Front.*, 10 (2019), pp. 1305-1326, 10.1016/j.gsf.2019.04.002
These authors claim mantle cyclicity of ca 93 and 187 Myrs.

Yes, now corrected.

Line 466: You present data, not evidence. Evidence is a very strong word choice. I would suggest rephrasing.

Agree. Replaced 'evidence' by 'data'.

Lines 480–488: Recent studies have shown, using several different basis of evidence that there is a lot of water in the mantle. Hence, I would strongly argue that subduction is NOT a prerequisite for the generation of silicic magma.

See for example papers below.

1. Schmandt, B., & Jacobsen, S. D. (2014). Dehydration melting at the top of the lower mantle. *Science*, 344(6189), 1259-1262.
2. Pearson, D. G., Brenker, F. E., Nestola, F., McNeill, J., Nasdala, L., Hutchison, B. J., ... & Stachel, T. (2014). Hydrous mantle transition zone indicated by ringwoodite from Brazilian diamond. *Nature*, 507(7491), 221-224.
3. Wang, W., Schmandt, B., & Hu, J. (2025, forthcoming or recent). Water Stored in the Mantle for Millions of Years May Be Linked to Continental Volcanism.
4. Bolfan-Casanova, N. (2018). Water in the Earth's mantle. *Mineralogical Magazine*, 69(3), 229-257.

In summary, the scientific consensus over the last decade has solidified the understanding that the Earth's mantle, particularly the transition zone, is a vast reservoir of water, likely holding several times the volume of our surface oceans. This understanding is built on a multidisciplinary approach combining seismology, high-pressure mineral physics, and geochemistry.

We also agree that subduction is not a prerequisite for the generation of silicic magmas. However, the upper mantle must first be hydrothermalized in order to produce basalts with arc-like geochemical signatures (e.g., Ernst, 2014; Pearce et al., 2021), such as negative Nb–Ta anomalies. This observation is an excellent contribution and further strengthens our interpretation; accordingly, we have rephrased the sentence as shown below.

‘Nonetheless, the mantle transition zone is a major H₂O reservoir (Schmandt et al., 2014), and hydrous upwellings from this region may generate intraplate volcanism (Wang et al., 2025). Large volumes of silicic magmas, however, can only be produced predominantly through partial melting of fertile, hydrous lower-crustal material previously generated by subduction (Bryan, 2007). Such processes not only facilitate the generation of extensive silicic magmatism but also account for mantle-normalized negative Nb and Ta anomalies, together with elevated Pb concentrations.’

Figure. 1A. The colour difference between the eclogite and ophiolites are too small. I would recommend changing these (or one of these) colours.

Ok, changed.

Figure. 2. I cannot find dyke swarm C2 in the figure.

We changed a little bit the red tone and added labels for the dyke swarms into the map in Fig 2A.

Figure 3C. Are the error bars 1 or 2 sigma?

1 sigma, now informed in the figure caption.

Figure 4. The authors mention second stage TDM ages. These are: 1. Not defined by ¹⁷⁶Lu/¹⁷⁷Hf for the second stage. 2. They are not presented anywhere.

These are second stage Nd TDM ages compiled from Motta et al. (2022) and Puetz and Condie (2021) present in the Supplementary Table 1.

Figure 5. Why are charnokites and 2.11–2.08 granitic magmatism denoted using magma blobs instead of squares like other magmatic events/intrusions, like for example the 2.0–1.93 Grnaultite, charnokite, high T/P gneisses?

Ok, blobs replaced by squares to match with the other events.

Figure 6. The evolution of the figures from A–D are not discussed. Please expand the figure caption to guide the reader what is happening in the transition between these stages.

Ok, now the capture is extended so A, B, C and D are properly explained.

Figure 7. The colour difference between the greenstone belts and cycle 1 is too small in (B).

Ok, changed to a lighter green color and enlarged the contour of Cycle 1 line a little bit.

Introduction

Lines 27-30: “The period from the Late-Rhyacian to the 27 Statherian includes anachronistic events of rift-drift, accretionary and collisional orogenies in most Paleoproterozoic belts, with episodes of silicic additions to the continental crust. An exception to these belts is the Late-Rhyacian to Statherian record of the Amazon Craton in South America.”

Why is the Amazon Craton record an exception? As you mention further down, in the geological context, the evolution of the Amazon Craton has long been interpreted as the result of southwestward continental growth, characterized by the continuous migration of tectonic and magmatic processes from an ancient Archean core in the east. This process is thought to have occurred through successive magmatic arcs accreted along the southwestern margin of the craton or the development of a long-lived accretionary belt (e.g., Tassinari and Macambira, 1999; Santos et al., 2000; Cordani and Teixeira, 2007; Scandolara et al., 2017; Trevisan et al., 2021; Motta et al., 2022), in line with the broader concept of the Great Proterozoic Accretionary Orogen (Condie, 2013).

Lines 31-33: This is a key area to investigate mechanisms of crustal growth and a potential relationship with subduction since **no plate tectonic indicators have been identified. Instead**, the period from 2.0 Ga Ma to 1.74 Ga Ma in this crustal segment is marked by the emplacement of large igneous provinces with an extensive bimodal igneous association.

This sentence likely needs rephrasing, as many will disagree with its current wording. What do the authors consider to be “plate tectonic indicators”? Arc-related geochemical signatures in Paleo- to Mesoproterozoic magmatic rocks, along with increasingly positive ϵ_{Nd} and ϵ_{Hf} values with decreasing U-Pb crystallization ages away from the craton are typical characteristics of accretionary orogens. Additionally, “instead” may not be the most appropriate transition. A more suitable alternative would be: “Between 2.0 and 1.74 Ga, the Amazon Craton also records the emplacement of large igneous provinces with extensive bimodal magmatism.”

I will return to this point later, but I believe a key aspect of the manuscript should focus on whether the presence of LIPs/SLIPs necessarily contradicts the widely accepted accretionary orogeny model for the evolution of the Amazon Craton. Alternatively, could these processes be reconciled? If so, craton growth could be understood as the result of both mechanisms—horizontal growth through accretionary processes and vertical growth associated with the emplacement of LIPs/SLIPs away from the craton margin (subduction zone). As noted by Teixeira et al. (2019), LIPs/SLIPs “appear to accompany the stepwise accretionary crustal growth of Amazonia,” and several other authors have linked them to slab retreat events.

Geological context of the Amazon Craton

General comment:

This section would greatly benefit from a simplified geological map of the Amazon Craton. It is extremely difficult to follow the text without a figure (and I am rather familiar with the Amazon Craton).

The figure should display: i) the main tectonic provinces of the Amazon Craton, with their ages indicated in the legend; the superposition of the studied SLIPs; iii) the locations of the analysed

samples. Additionally, since the "SAMBA" model is referenced, it would be useful to include it as an inset within the geological map.

Furthermore, given that you have a detailed map of the Guiana Shield (current Fig. 4), it would be appropriate to provide a similar one for the Central Brazil Shield, as the majority of the studied samples come from this area.

Lines 59-53: "The distribution of Archean crust is evidenced as an isotopic provinciality that divides the Amazon Craton into an eastern segment, characterized by Archean Nd depleted mantle model (TDM) ages and the most negative $\epsilon\text{Nd}(t)$ values, and a western segment, where Rhyacian to Orosirian Nd TDM ages and slightly negative to slightly positive $\epsilon\text{Nd}(t)$ values predominate (Fig. 1a, b)."

This figure does not seem to be the most appropriate choice for this section. A Nd TDM age map is not necessary to highlight the Archean crust and the Paleo- to Mesoproterozoic provinces; a geological map is much more effective. To enhance clarity, as mentioned earlier, the SLIPs and sample locations should be plotted on a geological map, rather than on an Nd TDM age or $\epsilon\text{Nd}(t)$ map.

Comments about Fig. 1:

"The database for contour construction is sourced from 27 and 21."

i) Reference '27', Condie et al., 2021, does not include any Nd isotopic data. Similarly, '21', Motta et al., 2022, only provides Nd data for the Central-Brazil Shield. Could the authors clarify the source of the remaining data?

ii) If the data is from different sources, did the authors confirm that the TDM ages were all calculated using the same Depleted Mantle model? If not, they should consider recalculating them for consistency.

iii) The Nd data used should be provided as supplementary material (figures must be reproducible by the reviewers and future readers). The figure should also include the sample positions so that the reader can assess the sample density and distribution.

iv) The software used and the interpolation method applied for generating the data in the figure should be specified.

Lines 75-79: "Nevertheless, the record of plate tectonic indicators in the Amazon Craton comprise a 1460-1440 Ma ophiolite in the Sunsás Belt, that post-dates Paleoproterozoic belts and pre-dates the late Mesoproterozoic events (1.2 – 0.95 Ga). These events are correlated with the Grenville-type orogens related to the collision of Amazonia and Laurentia".

I'm not sure why the authors included this or what they intend to convey with this text.

Lines 79-82: Further, the proposition of Orosirian-Statherian orogenic belts in the Amazon Craton does not match with some important surface geological relationships, including two Orosirian and one Statherian igneous belts exhibiting N-S and E-W, NW-SE and E-W trends that conflict with the parallel NW-SE orogenic provinces."

I don't fully understand the sentence. From the regional geological maps provided by other authors (since this manuscript does not include one), it is very clear that the provinces are generally oriented in a NW-SE direction. This orientation is well-documented and widely

accepted in the published literature. If the authors are suggesting a different orientation, they should provide evidence to support this claim.

Orosirian and Statherian igneous belts:

It is essential to provide at least a brief description of the geology of each of the igneous belts. Are there notable differences between them, or are they very similar? As it currently stands, without referring to the literature, all I know is that there are three Paleoproterozoic SLIPs in the Amazon Craton and their respective ages, nothing more. The lithologies they encompass, their relationships with one another, etc, remain unclear.

You analysed 15 volcanic rocks, mostly ignimbrites, from three SLIPs. While the supplementary material lists the geological unit of each sample, there is no context provided. The reader is left uncertain about the nature of these formations, whether they are associated with sedimentary rocks, how they relate to the calc-alkaline I-type to A-type plutonic rocks, and so on. There is a significant lack of geological context. In addition, there is an absence of field and petrographic photos of the samples. While the field constraints might be well-established, and I don't doubt that, this information is not conveyed in the manuscript.

Results

Age of volcanic rocks

Please refer to the comments in the separate Word file regarding the U-Pb data. Additionally, I believe this section would benefit from a figure comparing your data with previously published data from the literature.

Zircon composition, crystallization conditions and source-depth

The calculated Ti-in-zircon crystallization temperatures (which are not presented as a figure in the manuscript or as a supplementary figure) and ΔQFM values should be provided in the supplementary data table. Additionally, is there any correlation between these values and the U-Pb ages and $\epsilon Hf(i)$ values of the analysed zircons? Additional insights may be gained from this.

Lines 175-178: "To estimate crustal thickness for each studied magmatic cycle we applied the chemical mohometry model, by using a large dataset of geochemical data available from the literature. Data was filtered to exclude major and mobile trace elements sensitive to hydrothermal alteration processes."

Data should be presented as supplementary material as it must be available to the reviewers to ensure the calculations are correct.

Lines 180-184: KDE plots of the depth estimate proxy using the $^{176}Lu/^{177}Hf$ ratio are not presented as a figure in the manuscript or as a supplementary figure. Results could/should be presented in the supplementary data table.

Comment on Figures 2b and 2c: if I am not mistaken, the fields were created by the users. If this is the case, I believe the data compiled from the literature to construct the fields should be provided as supplementary data to allow the reviewers to verify the data and ensure the accuracy of the figures.

Hf isotopic signatures from Paleoproterozoic to 201 Ma in the Amazon Craton

Lines 201-204: "In addition to previously published datasets including detrital and igneous zircon age and Hf 203 isotopes, we present 172 coupled zircon U-Pb and Lu-Hf analyses from 15 volcanic rock samples representing the three Paleoproterozoic igneous cycles."

Reference 27, Condie et al. 2021, does not include any Hf isotopic data. Compiled Hf data should be provided as supplementary material and references indicated.

Lines 216-244: This part of the text focus on interpretation and implications rather than the presentation of the results, therefore, it would be more appropriate to move this content to the discussion section.

Comments of figure 3:

i) Detrital and igneous zircon Hf data from the literature are currently presented as grey circles. They should be represented with different symbols for clarity. Igneous zircon should be symbolized according to their geological unit or, alternatively, tectonic province to provide more context and facilitate interpretation.

ii) What exactly do the literature data for Cycle 1, Cycle 2, and Cycle 3 represent? Why are they separated from the general "detrital and igneous (literature)" data? Given that you only analysed volcanic rocks, mainly ignimbrites from each of the SLIPs, how do these volcanic rocks' Hf compositions compare to the plutonic rocks of each SLIP? Are there any notable differences or similarities between them that could provide additional insights into the geochemical characteristics or tectonic settings of these regions?

For example, zircon from the three analysed volcanic samples of the Uatamã SLIP (Cycle 2) show a relatively restricted range of $\epsilon_{\text{Hf}}(i)$ values, from -6.7 to +1.7, with an average value of -2.0. However, the "Cycle 2 literature" data shows a major cluster at -15.0. What lithologies does this cluster represent? Are they the same as the more radiogenic cluster or are they different? Why do the Uatamã SLIP rocks reach much more enriched (more negative) $\epsilon_{\text{Hf}}(i)$ values, approaching the evolution trend of the Archean crust as defined by the "detrital and igneous (literature)" zircon? (By the way, what exactly do these Archean "detrital and igneous (literature)" represent? Igneous rocks from Carajás? Archean detrital zircon from younger sequences?)

On another note, how do the Hf isotopic data from this study and the literature compilation compare with the Nd data from the literature? You compiled Nd data for the Amazon Craton, as shown in Fig. 1. Why not compare the two datasets? Do they show similar or divergent trends? This could provide further insights into the evolution of the Amazon Craton.

iii) It is unclear what the starting point ages are for the C-F2, TTG, and KREEP, and why those particular ages were chosen. Please specify the rationale behind these age selections.

iv) The references for the 'previously published zircon ages' used to construct the KDE diagram must be provided. Additionally, the compilation of age data, coordinates, and other relevant information should be included as supplementary material to ensure transparency and allow for reproducibility.

Discussion

Changes in structure and composition of the Amazon Craton crust

The discussion of the Hf data should be moved to this section, and should be discussed along with the Nd isotopic data.

Lines 292-293: "The isotope composition of silicic igneous rocks reveals that the proportion of reworked crust during magmatism was highly dependent on the availability of pre-existing crustal material."

While this statement might be valid for Cycle 3 rocks, as you explain in lines 296 to 299, it does not explain the differences observed between Cycle 2 and Cycle 1. Although there is no available projection of the Hf-analysed samples on a geological map, the existing projection on the Nd TDM age map suggests that both Cycle 1 and Cycle 2 rocks are spatially associated with Archean crust, whereas Cycle 3 rocks, further west, overlie a region characterized by predominantly Paleoproterozoic Nd TDM ages. If Cycle 1 and Cycle 2 broadly overlap spatially, their availability of pre-existing crustal material and its characteristics should be comparable. However, the isotopic results indicate that Cycle 2 rocks have incorporated significantly more evolved crustal material than Cycle 1.

Cycle 3 rocks exhibit an $\epsilon\text{Hf}(i)$ range identical to that of Cycle 1. Identical $\epsilon\text{Hf}(i)$ values for two magmatic events of distinct ages suggest either an increase in the juvenile component and/or a reduction in the influence of the Archean crust in the younger cycle. Given the apparent overall correspondence between the Nd and Hf isotopic data (as seen in both Fig. 1 of this study and Fig. 8 of Motta et al., 2022), and also that Cycle 3 rocks are located west of the Cycle 1 and 2 rocks (i.e., further away from the Archean core), the data strongly suggest that Cycle 3 rocks were emplaced in a distinct crustal "structure" (either younger crust or thinned Archean basement) compared to Cycles 1 and 2.

Comments on Figure 5: i) There are no KDEs shown on the right side of the figure. ii) The abbreviations MU, MC, UC, and SM are not indicated in the figure.

Repeated Cycles of Mantle-Derived Magmatism at ~95 Myr Intervals

Comments on Figure 6: i) Fig. 6A and B are exchanged. ii) Sources of data from the literature must be indicated in the figure caption. Compiled data should be made available as supplementary material.

Crustal growth by intraplate igneous events during the Paleoproterozoic

Lines 411-416: "Prior to this, from the Orosirian to the Statherian, no clear evidence of subduction events exists within the Amazon Craton following the Rhyacian Transamazonian-Eburnean-Birimian orogenies. Instead, crustal growth during this interval was primarily driven by large igneous province (LIP) events. A potential exception is the proposed Statherian arc magmatism in the Rio Negro Province; however, this interpretation remains unsupported by definitive plate tectonic indicators."

This is a classic case where the phrase "absence of evidence is not evidence of absence" applies. The absence of "definitive plate tectonic indicators" could be due to a true lack of subduction or

simply because of an incomplete record (ophiolite sequences and high-pressure rocks are not very common in the Paleoproterozoic).

The assertion that crustal growth during the Orosirian-Statherian was "primarily driven by LIP events" is a strong claim that requires robust evidence. While the authors correctly highlight the importance of LIPs in crustal growth, they do not sufficiently address the geochemical and isotopic trends that are commonly associated with accretionary orogens, such as the southwestward decrease in crystallization ages and TDM model ages observed in the magmatic rocks of the Paleoproterozoic Ventuari-Tapajós and Rio Negro-Juruena belts, which in addition exhibit arc-related geochemical signatures. This pattern is a hallmark of long-lived accretionary orogens from the Proterozoic to the Phanerozoic (e.g., Bahlburg et al., 2023 and references therein) and should not be dismissed without a compelling alternative explanation.

Instead of positioning LIP magmatism in opposition to accretionary tectonics, the authors should explore how these processes may have interacted. Could the emplacement of LIPs have played a role in modifying an accretionary orogen? Is there geochemical evidence suggesting interplay between plume-derived and arc-related magmatism? These are the types of questions that could provide a more nuanced and comprehensive interpretation of the Amazon Craton's evolution. A more balanced discussion would acknowledge the role of LIPs while evaluating whether they necessarily contradict an accretionary orogenic model. Rather than dismissing one process in favour of the other, the manuscript could explore how both may have contributed to crustal growth of the Amazon Craton.

Comments on the data and figures presented in the supplementary table:

1) When reviewing a manuscript with geochronology data, my first step is always to check the reference materials and to attempt reproducing the figures and age calculations. Results from the reference materials are in excellent agreement with the accepted values.

Using Isoplot-R and following the same conditions outlined in the Methods section, I tested the data reported in the supplementary table for the studied samples. While the calculated ages agree within uncertainty, **I cannot reproduce the age results and diagrams presented by the authors: the plots differ**. Below, I provide an example for sample VR05, though **this issue applies to all samples**. As shown, the calculated ages are consistent within uncertainty, but the plots themselves are different.

While this discrepancy does not affect the overall results of the manuscript, because the calculated ages usually overlap within uncertainty, it is important that any reader should be able to reproduce the figures presented by the authors. **This is an issue and should be addressed**.

I am not sure about the cause of these differences. One possibility is that the authors may have plotted the diagrams using the $^{207}\text{Pb}/^{235}\text{U}$ and $^{206}\text{Pb}/^{238}\text{U}$ ratios, rather than the $^{238}\text{U}/^{206}\text{Pb}$ and $^{207}\text{Pb}/^{206}\text{Pb}$ ratios reported in the table, meaning that the results calculated by Iolite are producing slightly inconsistent isotopic ratios (this is a known issue of Iolite). However, this is just a guess, as **the $^{207}\text{Pb}/^{235}\text{U}$ ratio is not included in the data table**, and I cannot confirm this. Depending on how the $^{207}\text{Pb}/^{235}\text{U}$ ratio was calculated (mean of ratios versus ratio of means), this could explain the discrepancies observed.

Authors should report the U-Pb isotopic data as follows:

$^{206}\text{Pb}/^{204}\text{Pb}$ (or % of common Pb), $^{207}\text{Pb}/^{235}\text{U}$, $^{206}\text{Pb}/^{238}\text{U}$, rho, $^{207}\text{Pb}/^{206}\text{Pb}$, $^{238}\text{U}/^{206}\text{Pb}$, rho, $^{232}\text{Th}/^{238}\text{U}$.

Plot for sample VR05 reported in supplementary table:

Plot for the same sample VR05 with data from columns B to F:

(Note that in the plot above, I did not reject the youngest and oldest analyses. To obtain an acceptable MSWD I would have had to reject more than 2.)

2) Furthermore:

- Please indicate the type of concordance used (e.g., % difference between the $^{206}\text{Pb}/^{238}\text{U}$ and $^{207}\text{Pb}/^{206}\text{Pb}$ ages, concordia distance, etc.). The authors appear to have used concordia distance (or a similar approach), but this should be explicitly stated.
- When examining the data for each individual sample, many of the rejected analyses could be interpreted as either significantly older and discordant grains (xenocrysts or inherited cores) or grains affected by common Pb. However, common Pb estimation is not provided in the data table, nor are CL images that could clarify whether these analyses correspond to inherited cores. **CL images of representative zircons of each sample should be provided.**
- Building on the above, when plotting the data for all samples, the common Pb hypothesis seems unlikely, as most data align along discordia lines toward geologically meaningful ages (those younger than 2.2 Ga). In contrast, analyses older than 2.2 Ga are more scattered and usually highly discordant (except for PB07A_8: 2852 Ma), making their $^{207}\text{Pb}/^{206}\text{Pb}$ ages questionable.

3) A potential approach for the treatment of U-Pb data could be considered. As noted by Vermeesch (2021, *Geoscience Frontiers* 12, 843–850), “volcanic rocks often exhibit positively skewed age distributions with a short tail of syn-eruptive U–Pb ages and a long tail of pre-eruptive or xenocrystic ages.”. In such cases, the maximum likelihood age model (minimum age on radialplot of Isoplot-R) might be a good alternative for determining the crystallization age, as it statistically rejects possible xenocrystic ages. This automated method is likely more accurate for identifying inherited grains than manually removing analyses by visual inspection. This is particularly relevant for Paleoproterozoic rocks, where a conservative 1% uncertainty corresponds to approximately ± 20 Ma, leading to inevitable overlap between crystallization ages and slightly older xenocrysts (e.g., see comments on sample GLR-35 below).

Some comments on the data of a few samples:

Sample LA-01:

- Grains LA01_17 and 18 overlap with the main group and, therefore, should not be considered as inherited. They should be included in the weighted mean $^{207}\text{Pb}/^{206}\text{Pb}$ and concordia age calculations. Based on the rejection criteria indicated in the supplementary table (concordance > 0.95), grain LA01_10 could also be included in the weighted mean $^{207}\text{Pb}/^{206}\text{Pb}$ age calculation.
- Grain LA01_19 (2064 ± 19 Ma) should be considered as an inherited grain as it aligns with LA01_3 (2061 ± 23 Ma) and 13 (2061 ± 22 Ma).
- Grains LA01_20, 24 and 25 seem affected by common Pb.

Sample PA-34:

The reported MSWD of 3.4 the weighted mean concordia age ($2002.3 \pm 3.2/6.3$ Ma) exceeds the range of acceptable MSWD values for $n = 19$ at 2 sigma level. The weighted mean $^{207}/^{206}$ age should be the preferred age (1996 ± 4 Ma, MSWD = 0.87).

Sample TM43:

Using the 15 analyses selected by the authors to calculate the weighted mean $^{207}\text{Pb}/^{206}\text{Pb}$ age, I obtain a lower MSWD value of 1.3, compared to the reported 2.5 in the data table. This suggests that the weighted mean $^{207}\text{Pb}/^{206}\text{Pb}$ age should be preferred over the weighted mean concordia age.

Sample MA-02:

Similarly to the sample above, using the 22 analyses selected by the authors to calculate the weighted mean $^{207}\text{Pb}/^{206}\text{Pb}$ age, I obtain a lower MSWD value of 0.56, compared to the reported 1.6 in the data table. In any case, both of these MSWD values are statistically more reliable than the MSWD of 4 reported for the weighted mean Concordia age, which was chosen as the preferred age by the authors. Given this, the weighted mean $^{207}\text{Pb}/^{206}\text{Pb}$ age should be preferred over the weighted mean Concordia age.

Sample GLR-35:

- The statistical parameters of the three “age” types provided clearly indicated that the calculated dates are not statistically sound:
 - concordia age (n = 23): MSWD = 6.8 and p = 0 (for concordance and equivalence);
 - weighted mean 207/206 age (n = 23): MSWD = 4.8, p = 0);
 - weighted mean concordia age (n = 20): MSWD = 3, p = 0).

The three analyses, considered inherited by the authors, are represented in grey. However, both the MSWD values and visual inspection of the diagram clearly indicate the presence of more than one population of analyses within the green group.

- Considering that the 23 analyses are all nearly concordant, we can use the radial plot function of Isoplot-R to assess the presence of more than one age component. The algorithm indicates the presence of two distinct age components within the 23 analyses, one at **1795 ± 6 Ma** and an older one at **1830.6 ± 8.9 Ma**. From the plot, it is clear that analyses older than 1820 Ma belong to the older component.
- Considering analyses older than 1820 Ma as inherited (GLR35_8, GLR35_16, GLR35_19, GLR35_22, GLR35_24, in addition to the 3 selected by the authors), we can calculate a weighted mean 207/206 age of **1795.2 ± 5.6 Ma** (MSWD = 0.82, p = 0.66) for the younger group.
- Using the MLA algorithm, we obtain an identical age of **1794 ± 7 Ma**.

Radial plot (finite mixtures = auto)

Radial plot (finite mixtures = minimum)

Sample PGR27:

- The authors consider only the existence of one inherited grain (PGR27_25). However, inspection of the Concordia diagram clearly indicates the presence of more inherited grains. Excluding analyses PGR27_8 and PGR27_9, which have very large uncertainties, it is clear that there are at least four analyses that are older than the main group.

- This is clearly depicted in the radial plot, which shows a younger population with a weighted mean $207/206$ age of 1791 ± 5 Ma and an older one at 1840.4 ± 9.9 Ma.

Sample SS-39:

Rejecting analyses SS39A_23 and SS39A_25 results in a statistically more robust concordia age (considering MSWD and p for concordance and equivalence).

Sample PB07A:

- Analyses PB07A_18 and PB07A_24 are not represented in the diagrams and were not used in the calculations; however, this is not indicated in the data table.
- From the inspection of the Concordia diagram, it is clear that, apart from grain PB07A_8 (2852 ± 17 Ma), there are other xenocrysts. This is also evident from the statistical parameters reported by the authors for the Concordia age, where MSWD = 5.1 and $p = 0$ (for concordance and equivalence) clearly point to excess scatter. Indeed, the radial plot indicates the presence of an older group comprising 4 analyses with a weighted mean 7/6 age of 1814 ± 12 Ma, and a younger group, the main one, at 1762.2 ± 5.4 Ma. Calculating a Concordia age for the latter group yields an excellent age of 1761.3 ± 3.9 Ma (MSWD = 1.1, $p = 0.28$, for concordance and equivalence).

Sample PB09A:

- This is a sample where the differences between the plots constructed for the same analyses with the data reported in the supplementary table (left) and the plot presented by the authors (right) differ significantly (see figures below). From the inspection of the diagram and the 7/6 ages (1810 ± 21 Ma and 1796 ± 22 Ma), it is clear that the two analyses in red on the left diagram are older than the main group (ages around 1760 Ma). This difference is not depicted in the diagram presented by the authors (right), which raises questions regarding the validity of the plots.
- As I mentioned before, while this does not have major implications for the calculation of the crystallization ages of the studied samples, since the final age differences overlap within uncertainty and therefore do not affect the conclusions of the study, it does indicate that something is wrong. While I can't be 100% sure that this is the cause of the issue, my guess is that it has to do with the fact that users plotted the diagrams using the 207/235 (not reported in the data table) and 206/238 ratios, meaning that the results calculated by Lolite are producing slightly inconsistent isotopic ratios (this is a known issue of Lolite).

- Considering the data presented in the data table, which is the one I have access to, this sample as an inherited component with a WM 7/6 age of **1803 ± 16 Ma** and a crystallization age of **1758.6 ± 5.4 Ma** (WM 7/6 age of the youngest group); the latter overlap with the Concordia age of 1758 ± 4 Ma (MSWD = 1.4, $p = 0.07$; excluding the most discordant analysis PB09_21).